# Gating of TonB-dependent transporters by substrate-specific forced remodelling

Samuel J. Hickman[1,2], Rachael E.M. Cooper[1], Luca Bellucci[3], Emanuele Paci[1,2] & David J. Brockwell[1,2]

Membrane proteins play vital roles in inside-out and outside-in signal transduction by responding to inputs that include mechanical stimuli. Mechanical gating may be mediated by the membrane or by protein(s) but evidence for the latter is scarce. Here we use force spectroscopy, protein engineering and bacterial growth assays to investigate the effects of force on complexes formed between TonB and TonB-dependent transporters (TBDT) from Gram-negative bacteria. We confirm the feasibility of protein-only mediated mechanical gating by demonstrating that the interaction between TonB and BtuB (a TBDT) is sufficiently strong under force to create a channel through the TBDT. In addition, by comparing the dimensions of the force-induced channel in BtuB and a second TBDT (FhuA), we show that the mechanical properties of the interaction are perfectly tuned to their function by inducing formation of a channel whose dimensions are tailored to the ligand.

[1] School of Molecular and Cellular Biology, Faculty of Biological Sciences, University of Leeds, Leeds LS2 9JT, UK. [2] Astbury Centre for Structural Molecular Biology, University of Leeds, Leeds LS2 9JT, UK. [3] NEST, Istituto Nanoscienze-CNR, Piazza San Silvestro, 12-56127 Pisa, Italy. Correspondence and requests for materials should be addressed to D.J.B. (email: d.j.brockwell@leeds.ac.uk).

Belying its name, the outer-membrane (OM) envelope of Gram-negative bacteria is relatively devoid of lipid and instead is packed with β-barrel OM proteins (OMPs), which function as enzymes, foldases, assembly platforms and both specific and non-specific transporters[1]. Many of these functions, such as transport against a concentration gradient, are energy dependent and, as the periplasmic space is devoid of adenosine triphosphate (ATP), it is necessary to couple OMP transporters to machinery from the energized inner-membrane (IM) to facilitate these processes. A well-studied example is the interaction of OM transporters with TonB—a periplasmic protein tethered to the IM. The TonB-dependent transporter family (TBDTs)[2–4] bind and then transport scarce but vital nutrients such as maltodextrin (MalA), sucrose (SuxA) and metallo-organic compounds including vitamin $B_{12}$ (BtuB), haem (HasR and HemR) and siderophores (FepA, FhuA, FecA and FhuE)[3,5]. Their importance to cell viability results in TBDTs being virulence factors in pathogenic bacteria[6,7]. Coincidentally, TBDTs are also hijacked by colicins and phage for cell entry[8].

Structurally, TBDTs are characterized by 22-stranded β-barrels whose lumens are occluded by an N-terminal globular 'plug' domain (Fig. 1a). The plug domain contains a conserved binding motif known as the Ton box[9] on the periplasmic side, which upon binding of extracellular substrate becomes disordered[10,11]. This transition allows the formation of a non-covalent complex with the C-terminal globular periplasmic domain of TonB ($TonB_{CTD}$) (Fig. 1b)[12,13]. TonB is anchored to the IM by a single N-terminal transmembrane helix and these domains are linked by a central proline-rich periplasmic spanning domain. This linker adopts an extended polyproline type II helical rod conformation[14], allowing $TonB_{CTD}$ to reach the OM. In vivo, TonB forms a complex with the IM proteins ExbB and ExbD[15]. These proteins are thought to function as a scaffold (ExbB)[16,17] and to harness the proton motive force (ExbD)[18], a pre-requisite for TonB-dependent activity.

While it is known that TonB-mediated transport requires TonB to be tethered to an energized IM[19], the mechanism by which TonB remodels the plug domains of TBDTs remains unclear[20]. Currently favoured models, such as the pulling hypothesis[21,22] or the rotational surveillance and energy transfer[23] model, suggest that TonB applies a mechanical remodelling force driven by its interaction with the ExbBD complex in the energized IM. This induces partial or full-plug domain unfolding via the non-covalent interaction of the Ton box with $TonB_{CTD}$[3,24–26]. For these models to be viable, the Ton box–TonB inter-protein interaction must be sufficiently stable under tension to allow the unfolding of the plug domain before its dissociation. Intriguingly, the crystal structure of BtuB (the vitamin $B_{12}$ TBDT receptor of E. coli) in complex with TonB shows that the Ton box binds to the β-sheet of $TonB_{CTD}$ in a parallel orientation and that this augmented β-sheet is rotated ∼90° with respect to the β-strands of the plug domain[12]. If the plug domain of BtuB is extended by TonB in vivo, the relative geometry of the β-strands involved in these inter- and

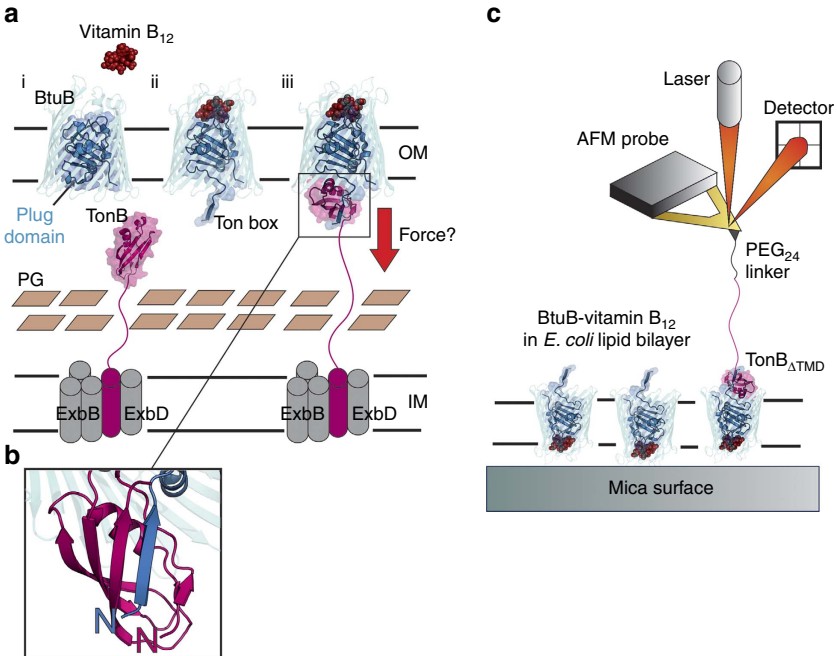

**Figure 1 | Investigating TonB-dependent transport by single-molecule force spectroscopy.** (**a**) Schematic of TonB-dependent vitamin $B_{12}$ transport in E. coli. PG, peptidoglycan. (i) The lumen of BtuB, a 22-stranded β-barrel OM protein (PDB: 1NQE), is occluded by an N-terminal plug domain (arrow) preventing transit of vitamin $B_{12}$ (red space filling model), (ii) the binding of vitamin $B_{12}$ induces an allosteric rearrangement of the plug domain, releasing the Ton box into the periplasmic space, where it forms a 1:1 complex (PDB: 2GSK) with the C-terminal domain of TonB ($TonB_{CTD}$, residues 153–233 (PDB: 1XX3). The transmembrane helix at the N-terminus of TonB forms a complex with ExbB and ExbD (grey cylinders), which is necessary, together with an energized IM, for TonB-dependent transport, (iii) linkage of the OM and IM via this non-covalent complex is thought to trigger the full or partial unfolding of the plug domain, allowing the passage of vitamin $B_{12}$. While the precise mechanisms vary, most models for TonB-dependent transport suggest that remodelling is induced by application of force (red arrow). (**b**) Detail of the TonB:BtuB interaction showing the parallel orientation of the β-strand augmentation interaction of the Ton box of BtuB (blue) with $TonB_{CTD}$ (pink). Note: for $TonB_{CTD}$, N designates the start of the C-terminal domain. (**c**) Simplified schematic to describe the single-molecule force spectroscopy approach used in this study. The interaction between soluble TonB (residues 33–239, $TonB_{\Delta TMD}$, that is, deletion of the N-terminal transmembrane helix) attached to the AFM probe and BtuB inserted into E. coli polar lipid extract liposomes and adsorbed onto a mica surface, is measured using an AFM. Note: adsorption of liposomes onto mica generates a heterogeneous surface with regions of single- and double-bilayer thickness.

intra-protein interactions are ideally oriented to engender these mechanical phenotypes[27,28] as shown using molecular dynamics (MD)[22].

To investigate the mechanism of TBDT function and more generally, the role of protein–protein interactions in membrane protein gating, we first employ single-molecule force spectroscopy using an atomic force microscope (AFM) to show that the non-covalent interaction between the Ton box from BtuB (TB$_{BtuB}$) and TonB is surprisingly durable under extension. Repeating this experiment using BtuB reconstituted in *E. coli* polar liposomes adsorbed onto a mica surface and TonB immobilized onto an AFM tip (Fig. 1c), we find that the TB$_{BtuB}$–TonB complex is sufficiently durable under force to allow the unfolding of half of the plug domain before its dissociation. The presence and extent of the mechanically labile subdomain is confirmed by repeating the experiments on disulfide crosslinked variants and the consequences of preventing plug remodelling is assessed using a vitamin B$_{12}$ transport assay. Finally, we show using mutagenesis that destabilization of the mechanically strong plug subdomain results in complete plug unfolding under force before dissociation of TonB and use an antibiotic sensitivity assay to show that this also occurs *in vivo*. Together, these data provide evidence for the direct mechanical gating of a membrane protein via protein–protein interactions and show that TBDTs maintain the integrity of the OM during transport by formation of a channel whose dimensions matches that of their cargo.

## Results

**TB$_{BtuB}$ interaction with TonB is mechanically strong.** The BtuB–TonB complex is thought to span the periplasm and, ignoring the membrane-embedded components, comprises three mechanical elements: the plug domain of BtuB, the TB$_{BtuB}$–TonB linkage and the polyproline linker domain of TonB. As a first step towards understanding the effects of mechanical extension on this network, we focussed on the non-covalent TonB–TB$_{BtuB}$ interaction using a minimal system comprising TonB$_{\Delta TMD}$ (TonB without the transmembrane domain (residues 1–32, see 'Methods' section and Fig. 2a) and a synthetic TB$_{BtuB}$ peptide (residues 5–14 of BtuB, see 'Methods' section). Before AFM experiments, the affinity of TB$_{BtuB}$–TonB$_{\Delta TMD}$ was measured by microscale thermophoresis yielding a $K_D$ of 9.4 ± 0.3 µM (Fig. 2b), similar to that previously published (36 µM) for the interaction between the Ton box peptide of FhuA and TonB$_{CTD}$[29]. TB$_{BtuB}$ and TonB$_{\Delta TMD}$ were then immobilized to an AFM probe and a silicon nitride surface, respectively, using flexible NHS-PEG$_{24}$-maleimide linkers (see 'Methods' section, Fig. 2a). The mechanical strength of this complex was then quantified by accumulating data from multiple approach-retract cycles whereby complex formation was allowed to occur in buffered solution by approaching the AFM probe to the surface at 1 µm s$^{-1}$ and, after application of a 300 pN trigger force, retracting at a velocity of 1 µm s$^{-1}$. Cycles that result in formation of the complex (7.5%) are identified by a distinctive single 'saw-tooth' force–extension profile (red traces Fig. 2c, left) resulting from extension of the linkers and the complex and subsequent dissociation of TB$_{BtuB}$–TonB$_{\Delta TMD}$ at a certain force. The rising edge of such profiles can be fitted with a worm-like chain model[30] (WLC, black solid lines Fig. 2c left, 'Methods' section) to obtain the maximum end-to-end length (or contour length $L_c$) of the force-resistant structure that comprises the protein complex and linkers at rupture. Data from ∼400 force–extension profiles (per experiment) are used to generate force- and contour length-frequency histograms (see Supplementary Fig. 1), allowing quantification of the most probable unbinding force ($F_U$) and most probable $L_c$ at rupture. Finally, as both the $F_U$ and $L_c$ for the unbinding of a specific interaction would be

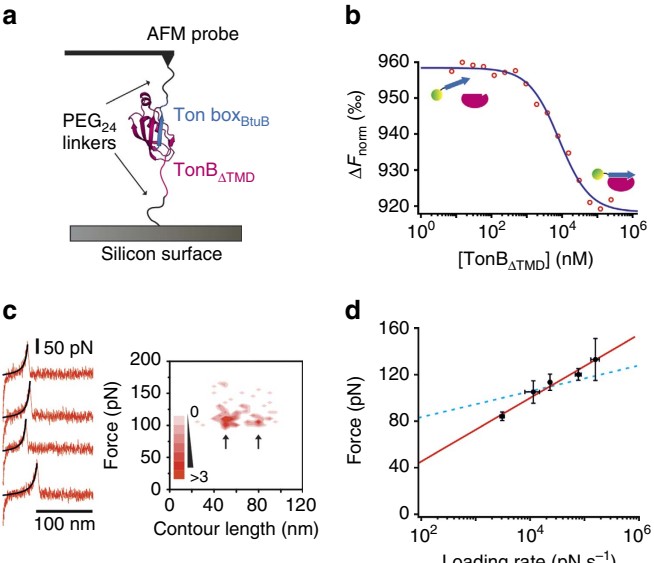

**Figure 2 | Characterization of the Ton box$_{BtuB}$ peptide:TonB$_{\Delta TMD}$ interaction.** (**a**) Cartoon schematic of the AFM experiment used to characterize the mechanical strength of TB$_{BtuB}$:TonB$_{\Delta TMD}$. (**b**) MST of the Ton box of BtuB (TB$_{BtuB}$) labelled with Alexa Fluor-488 C5 maleimide in the presence of 7 nM-244 µM TonB$_{\Delta TMD}$. Data (open red circles) are fitted using the law of mass action with the Nanotemper software to give a $K_D$ of 9.4 ± 0.3 µM. (**c**) Left: example retraction traces (red) showing the extension and rupture of TB$_{BtuB}$:TonB$_{\Delta TMD}$ at a retraction velocity of 1 µm s$^{-1}$ fitted with a worm-like chain model (WLC, black). Right: a contour length ($L_c$) versus rupture force ($F_U$) scatterplot of 90 dissociation events reveals two distinct hot-spots (arrows). Scatterplot is coloured by density of data points (5 nm-10 pN). (**d**) The dynamic force spectrum (a plot of $F_U$ versus the natural logarithm of the force loading rate (pN s$^{-1}$)) of the TB$_{BtuB}$:TonB$_{\Delta TMD}$ interaction. The error bars show the range of measurements from the triplicate mean. Data are fitted to a linear regression (red line). The dynamic force spectrum obtained for the highly avid E9:Im9 (E9 crosslinked between residues 20–66) is shown for comparison (dashed blue line)[33].

expected to be both correlated and reproducible, combining these data in a scatterplot allows the specificity of the interaction to be assessed by the presence or absence of a 'hot-spot'. The specificity of the data presented here is evidenced by comparing the scatterplots obtained for TonB$_{\Delta TMD}$ with wild-type TB$_{BtuB}$ (hotspot present, see Fig. 2c, right and Supplementary Fig. 2a) and a binding deficient variant of TB$_{BtuB}$[31] that substitutes a proline for leucine at position 8 (no hotspot, Supplementary Fig. 2a).

Closer examination of the individual force–extension profiles (Fig. 2c, left), distance histograms (Supplementary Fig. 1b) and scatterplots (Supplementary Fig. 2a) obtained over a range of retraction velocities (0.2–5 µm s$^{-1}$) reveal that while dissociation occurs at a single force ($F_U = 113 \pm 13$ pN at 1 µm s$^{-1}$ (average of the mode of the Gaussian fit of triplicate data sets ± half of the range), two $L_c$ distributions are evident with modal values of 47 and 74 nm. The latter value is consistent with the sum of the end-to-end length of the PEG$_{24}$ linkers (19 nm) and the expected end-to-end length of the TonB$_{\Delta TMD}$–TB$_{BtuB}$ complex (5.5 nm) with an unstructured or unfolded 118-residue linker domain (47.2 nm, see 'Methods' section and Supplementary Fig. 2b). The shorter modal $L_c$ value (coincidentally also 47 nm) suggests that the linker domain of TonB$_{\Delta TMD}$ must also populate a force-resistant conformation ∼22.5 nm in length (47–19–5.5 nm). To test this hypothesis, these experiments were repeated using

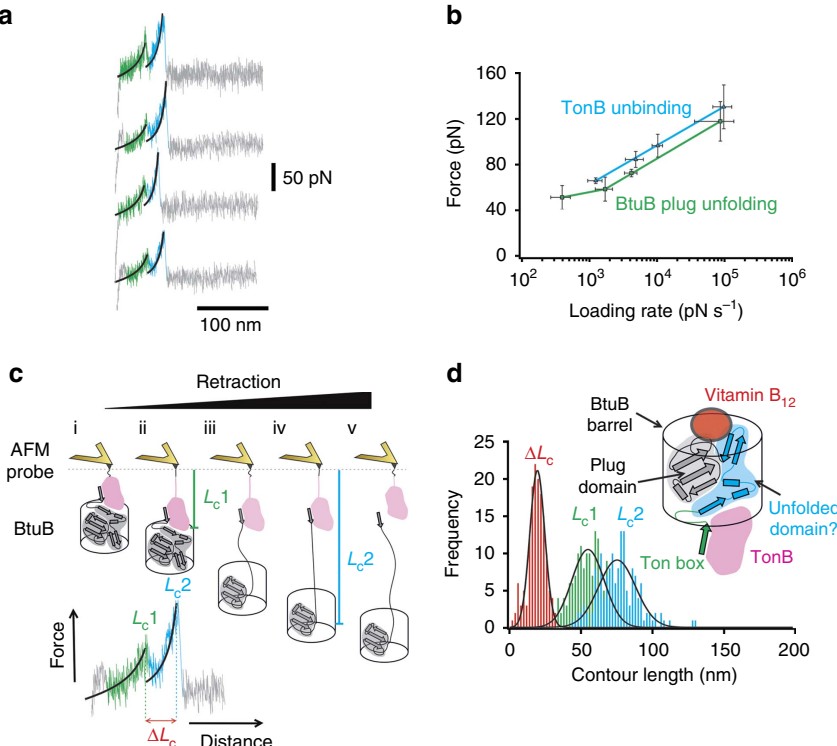

**Figure 3 | Pulling on the BtuB–TonB complex partially unfolds the plug domain. (a)** Representative force–extension profiles that show two rupture events (the leading edge of the first and second event is coloured green and blue, respectively). Each is fitted to the WLC model (black). **(b)** Dynamic force spectrum of unfolding (green) and unbinding (blue) events at 200, 500, 1,000 and 5,000 nm s$^{-1}$, error bars show average and range of fitted mode of three triplicate data sets. **(c)** Schematic describing the origin of each rupture event (an example trace with WLC fits is shown below) and their relationship to $L_{c1}$, $L_{c2}$ and $\Delta L_c$. The barrel of BtuB is shown as an open cylinder with the secondary structure of the plug domain (grey shaded region) and TonB (pink), which is attached to the AFM probe. (i) The AFM probe is pressed against the surface bringing the TonB–BtuB complex together; (ii) TonB bound to BtuB is fully extended causing the force to increase; (iii) part of the plug domain is unfolded and the force sharply decreases; (iv) the unfolded region is extended, causing an increase in the entropic restoring force and (v) TB$_{BtuB}$ dissociates from TonB and the force sharply falls to zero. The difference in fitted contour lengths ($\Delta L_c$) between the two peaks can be used to calculate the extent of unfolding. **(d)** Contour length frequency distributions from WLC fits to double-peaked force–extension profiles from a single dataset ($n = 153$) accumulated at a retraction velocity 1,000 nm s$^{-1}$. Frequency distributions obtained for the remodelling/unfolding event ($L_{c1}$, green), unbinding ($L_{c2}$, blue) and $\Delta L_c$ (red) are fitted to Gaussian functions (black lines). Inset: the presence of an unfolding event before dissociation reveals that the plug domain comprises a mechanically weak subdomain (the 50 amino acids downstream of the Ton box (residues 23–73, shaded blue)) and a mechanically recalcitrant subdomain (shaded grey). Residues that extend before the first event are coloured green. TB$_{BtuB}$ is shown as a green arrow bound to TonB (shaded in pink).

TonB$_{CTD}$ (that is, the globular C-terminal domain without the linker). These experiments yielded a unimodal distance-frequency histogram with a Gaussian fit value (22.4 nm) close to the predicted value (24.5 nm, the sum of the PEG$_{24}$ linkers (19 nm) and TonB$_{CTD}$–TB$_{BtuB}$ complex (5.5 nm), Supplementary Fig. 2c).

As forced-dissociation experiments are typically performed far from equilibrium, $F_U$ depends on the rate at which force is loaded onto proteins and their complexes[32]. To quantify the loading rate dependence of this system, TonB$_{\Delta TMD}$–TB$_{BtuB}$ was extended at retraction velocities of 200–5,000 nm s$^{-1}$ (corresponding to force loading rates of $10^3$–$10^5$ pN s$^{-1}$ (see 'Methods' section). The resultant dynamic force spectrum reveals that rupture occurs at 84–132 pN (Fig. 2d) at the relatively high-loading rates applied using AFM. This is a remarkably strong interaction for a complex with a relatively low affinity (µM) if the association rate is assumed to be diffusion limited. For example, at similar loading rates, the dissociation of E9$_{20-66}$-Im9 ($K_d = 10^{-14}$ M, $k_{off} = 1.4 \times 10^{-6}$ s$^{-1}$)[33] and an antibody and its epitope ($K_d = 10^{-9}$ M, $k_{off} = 4.4 \times 10^{-3}$ s$^{-1}$)[34] occurs at a force of $\sim 100$ pN and $\sim 160$ pN, respectively. These comparisons indicate that the TonB–Ton box complex is indeed mechanically robust and

provide support for a slow association rate[29]. It is interesting to note that the β-strand augmentation mechanism which underlies complex formation is well known for being mechanically 'strong'[12] and is found in other complexes evolved to withstand mechanical stress[28].

**Force-induced partial unfolding of BtuB by TonB$_{\Delta TMD}$.** We have demonstrated above that TonB$_{\Delta TMD}$–TB$_{BtuB}$ is able to resist levels of force that are sufficient to unfold protein domains with moderate mechanical strength[35,36] at similar applied loading rates. This raised the possibility that the plug domain, which is contiguous with the Ton box, may fully or partially unfold to open the lumen before dissociation of TonB from BtuB. To test this theory, wild-type BtuB was overexpressed and purified from the OM of *E. coli* and reconstituted into liposomes composed of *E. coli* polar lipid extract in the presence of cyanocobalamin (a synthetic form of vitamin B$_{12}$) (see 'Methods' section). The lipid vesicles containing BtuB were then adsorbed onto a freshly cleaved mica surface and TonB$_{\Delta TMD}$ immobilized to the AFM probe (Fig. 1c). Pressing the probe against the surface with 60–90 pN of force and withdrawing at a velocity of 1 µm s$^{-1}$

(force loading rate $\sim 10\,\mathrm{nN\,s^{-1}}$) after dwelling for 1 s led to two distinctive force–extension profiles. Approximately a third of the profiles (31%) displayed a single rupture event ($F_U = 97 \pm 8\,\mathrm{pN}$, $L_c = 51 \pm 3\,\mathrm{nm}$) similar in force to that observed for the minimal TonB–TB$_{BtuB}$ interaction at the same loading rate ($F_U = 113 \pm 7\,\mathrm{pN}$). The majority, however, displayed two rupture events (Fig. 3a) at $F_U$ and $L_c$ values of $61 \pm 4$ and $91 \pm 23\,\mathrm{pN}$ and $58 \pm 3$ and $77 \pm 7\,\mathrm{nm}$, respectively (Supplementary Table 1). These more frequent events indicate that the complex must undergo some partial unfolding before dissociation and that unfolding is more likely than dissociation of the complex at this loading rate. As we have shown that TonB does not unfold at these forces, force would presumably be propagated into the plug domain via the contiguous Ton box. These data thus suggest that the strength of intra-polypeptide interactions in the plug domain are weaker than those of the inter-polypeptide interactions between TonB$_{\Delta TMD}$ and BtuB, leading to partial/full unfolding of the plug before unbinding. As the mechanical strength of proteins and their complexes is kinetically controlled, the loading rate may affect unfolding and unbinding differently. Interestingly, analysis of the speed dependence of the putative unfolding and unbinding events demonstrates that the relative strength of intra- and inter-protein interactions at the same loading rate remains constant across the dynamic range of experiment (Fig. 3b).

**Identifying a force-induced channel through the plug domain.** To quantify the extent of unfolding, force–extension profiles showing two events were analysed further by fitting worm-like chain models to the unfolding ($L_{c1}$) and unbinding ($L_{c2}$) events. The difference in these values ($\Delta L_c$) reveals how much of the

protein complex becomes unfolded at the first rupture event (Fig. 3c). These data in the form of a histogram are shown in Fig. 3d and reveal a modal value of $\Delta L_c = 20 \pm 2\,\mathrm{nm}$ that is independent of the retraction velocity used (Supplementary Table 1), or importantly, the presence or absence of the TonB linker domain (Supplementary Fig. 3 and Supplementary Table 2). This extension corresponds to the unfolding of 50 amino acids—approximately half of the residues within the plug domain. While these data reveal the extent of unfolding it does not pinpoint the region involved within the plug. To do this, we reasoned that the 50 residues directly downstream of the Ton box motif (residues 23–73) would most likely unfold first (see Fig. 3c,d, inset), as these residues are initially exposed to mechanical deformation as force propagates through the plug. To test this assumption, BtuB variants that contained a disulfide bridge within the plug were generated. First, a variant designated XL$_{barrel}$ (Fig. 4a schematic) was designed to prevent any unfolding by linking the N-terminus of the plug domain (residue 23 located between the Ton box and the first β-strand of the plug domain) and residue 374 in strand 12 of the barrel domain (BtuB L23C/S374C, yellow-filled circles joined by red bar, Fig. 4a, left). For this crosslinked variant, 74% of the force–extension profiles analysed contained single rupture events with an $F_U$ of $100 \pm 6\,\mathrm{pN}$ at a $L_c$ of $47 \pm 2\,\mathrm{nm}$ (Fig. 4b, red traces and histogram). The $L_c$ value is in accord with the distance expected for the rupture of the complex without remodelling, while the $F_U$ value is close to that observed for the dissociation of the minimal TB$_{BtuB}$–TonB complex observed previously at the same retraction velocity ($1\,\mathrm{\mu m\,s^{-1}}$, $113 \pm 13\,\mathrm{pN}$). Addition of reductant ($2\,\mathrm{mM}$ dithiothreitol, DTT) to the buffer solution during the AFM experiments

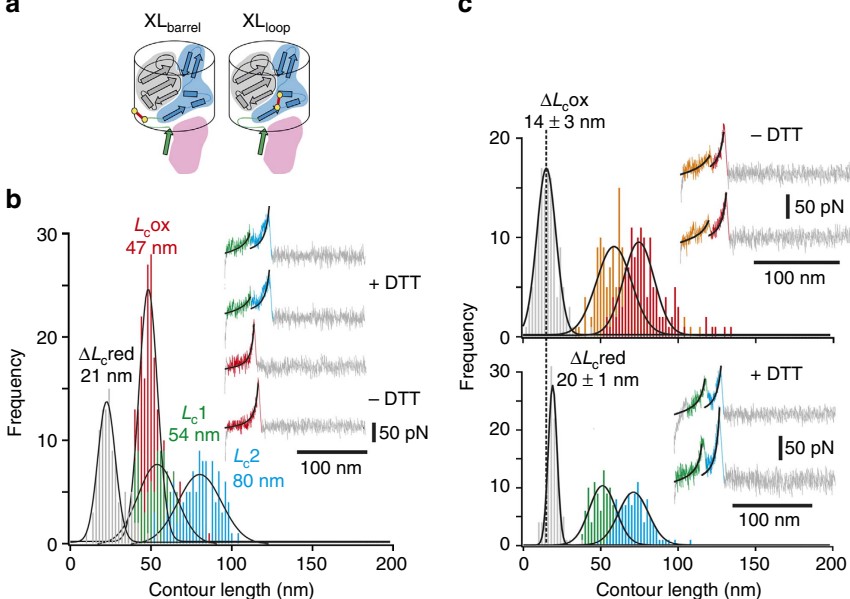

**Figure 4 | Using disulfide crosslinking to locate the mechanically weak plug subdomain.** (**a**) Secondary structure and topology of the plug domain showing location of the disulfide bridges (bold red line linked by yellow circles). The mechanically weak and strong subdomains are shaded blue and grey, respectively, and TB$_{BtuB}$ is shown as a green arrow bound to TonB (shaded in pink). (**b**) $L_c$ distributions of single rupture events observed for BtuB XL$_{barrel}$–TonB$_{\Delta TMD}$ in the absence of reductant ($n = 196$, red histogram, red force–extension traces (inset)) and, after addition of $2\,\mathrm{mM}$ DTT ($n = 109$), double rupture events resulting from partial unfolding ($L_{c1}$, green histogram and green force–extension profiles, inset) and unbinding ($L_{c2}$, blue histogram and blue force–extension profiles, inset) of XL$_{barrel}$ plug domain from TB$_{BtuB}$. When fitted to a Gaussian distribution, the $\Delta L_c$-frequency histogram (grey bars) for these double events has a modal value of $21 \pm 4\,\mathrm{nm}$. (**c**) Top: $L_c$ distributions and force–extension profiles (inset) for BtuB XL$_{loop}$–TonB$_{\Delta TMD}$ in the absence of DTT ($n = 154$). Partial unfolding ($L_{c1}$) and unbinding ($L_{c2}$) are shown in orange and red, respectively, for both histograms and force–extension profiles. The $\Delta L_c$ (grey histogram) modal value is shown by the dashed line. Bottom: $L_c$ distributions and force–extension profiles (inset) for BtuB XL$_{loop}$–TonB$_{\Delta TMD}$ in the presence of $2\,\mathrm{mM}$ DTT ($n = 122$). Partial unfolding ($L_{c1}$) and unbinding ($L_{c2}$) are shown in green and blue, respectively, for both histograms and force–extension profiles. In the presence of DTT, the $\Delta L_c$ (grey histogram) modal value ($20 \pm 1$) reverts to that of wild-type BtuB ($20 \pm 2$). All data collected at $1,000\,\mathrm{nm\,s^{-1}}$.

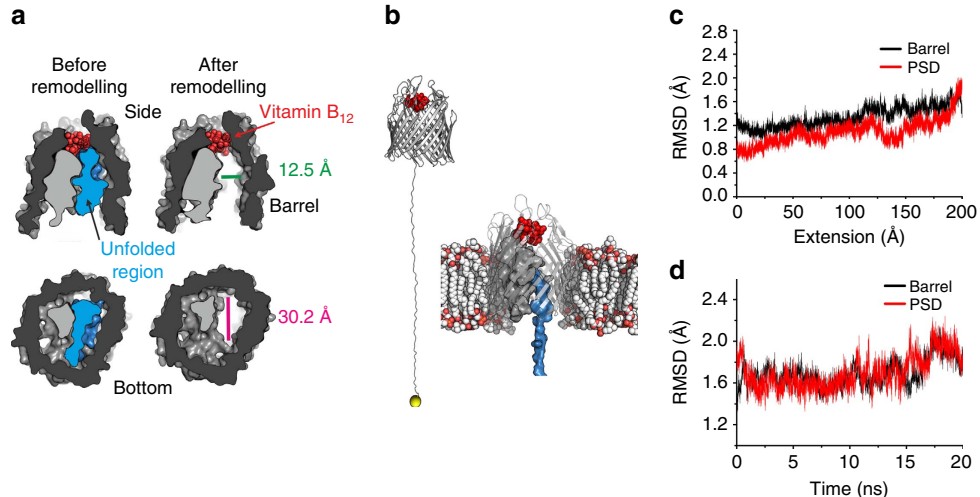

**Figure 5 | Creation of a channel through the receptor by partial unfolding of the plug domain.** (**a**) Surface representation of vertical and horizontal cross-sections of BtuB showing the effect of removing 50 amino acids (blue) downstream of the Ton box. The dimensions of the resultant channel are shown. (**b**) Left: snapshot after 20 nm SMD from the Ton box (yellow sphere), the membrane and water are hidden for clarity. BtuB is shown as a grey cartoon with vitamin $B_{12}$ shown as red spheres. Right: snapshot of BtuB equilibrated in a POPC bilayer (red and white spheres) after 20 nm of plug unfolding induced by extension of the Ton box by SMD. The unfolded mechanically weak domain is shown in blue surface representation, the recalcitrant plug subdomain (PSD) is shown in grey surface representation and vitamin $B_{12}$ as red spheres. (**c**) Root mean square deviation (a measure of the average deviation of the atomic positions from the experimental structure) of the recalcitrant PSD and barrel domain during 20 nm of SMD. The simulations show that the root mean square deviation of the recalcitrant PSD and the β-barrel are stable and of similar magnitude, demonstrating that the mechanically strong PSD is conformationally stable over this timescale. (**d**) Equilibration of the system after 20 nm of unfolding by SMD.

**Table 1 | Methionine assay to assess the growth phenotype of the BtuB variants.**

| Strain | Growth on: | | | | |
|---|---|---|---|---|---|
| | | **[Vitamin $B_{12}$] (nM)** | | | |
| | Met | 0.1 | 1 | 10 | 100 |
| RK5016 | | | | | |
| pAG1 (wt BtuB) | + + | + + | + + | + + | + + |
| L8P | + + | − | − | + | + |
| $XL_{barrel}$ | + + | + | + | + | + |
| $XL_{loop}$ | + + | + + | + + | + + | + + |

Growth of *E. coli* RK5016 *metE btub* cells transformed with BtuB variants after 48 h incubation on Davis minimal agar supplemented with 0.1–100 nM vitamin $B_{12}$ or 5 μg ml$^{-1}$ methionine. − indicates no observable colonies, + indicates partial growth (small colonies) and + + indicates full growth (comparable to methionine plate controls (colony diameter = 2 mm)). Two biological replicates were performed.

resulted in the re-appearance of the double-peaked events observed for the wild-type receptor (Fig. 4b, green and blue traces and histogram) with a modal $\Delta L_c$ value of $21 \pm 4$ nm (Fig. 4b, grey histogram) and $F_U = 64 \pm 1$ and $99 \pm 7$ pN for the unfolding and unbinding forces, respectively (Supplementary Table 3). To localize the unfolding event further, a V29C/V45C variant ($XL_{loop}$) was designed to crosslink strand 1 (V29C) and helix 2 (V45C) of the plug domain (yellow-filled circles joined by red bar, Fig. 4a, right). Upon disulfide bond formation, this creates a covalent 15 amino-acid loop within the region predicted to unfold (blue-shaded area in Fig. 4a schematic) shortening $\Delta L_c$ by ∼6 nm. Similar to the wild-type data, force–extension profiles for BtuB $XL_{loop}$–TonB dissociation contained a high proportion (94%) of double-peaked events (Fig. 4c, inset). Most importantly, the $\Delta L_c$-frequency histogram (Fig. 4c, grey histogram) yielded a modal value 6 nm shorter relative to wild-type ($14 \pm 3$ and $20 \pm 1$ nm), confirming unfolding in the N-terminal portion of the plug. Once again, the force–extension profiles for BtuB $XL_{loop}$–TonB dissociation reverted to wild-type profiles upon addition of 2 mM DTT ($\Delta L_c = 20 \pm 1$ nm) (Fig. 4c, bottom grey histogram). Together these data suggest that the first ∼50 residues of the plug domain are denatured before the non-

covalent TonB–BtuB complex dissociates. The structural consequence of partially unfolding the plug domain was visualized by removing the 50 residues directly downstream of $TB_{BtuB}$ from the crystal structure of BtuB bound to vitamin $B_{12}$ and $TonB_{CTD}$[12]. The topology of the plug domain is such that this deletion results in the appearance of $12.5 \times 30.2$ Å-wide channel through the receptor (Fig. 5a). To assess the effect of unfolding half of the plug domain, we performed MD simulations of the system. These simulations were similar to that undertaken by Gumbart and colleagues in 2007 apart from the presence of bound vitamin $B_{12}$ in this study (Supplementary Fig. 4). After the system was equilibrated (Supplementary Fig. 4), a force ramp was applied by retracting a harmonic spring at constant velocity (also sometimes called steered MD) until the Ton box was extended 20 nm away from its original position (Fig. 5b, left). Similar to the previous MD study on BtuB–TonB[22] (and also observed for the *Neisseria* TbpA–TonB complex[37]), the plug domain unfolded directly downstream of the Ton box creating a continuous channel through the receptor (Fig. 5b, right). Furthermore, the mechanically strong subdomain was found to remain native-like during (Fig. 5c) and up to 20 ns after unfolding (Fig. 5d) of the mechanically weak subdomain.

The effect of preventing or diverting the force propagation pathway through the plug domain *in vivo* was next assessed by examining the growth phenotypes of an *E. coli metE* strain expressing wild-type BtuB or its variants in the presence of different concentrations of vitamin B$_{12}$. This strain requires vitamin B$_{12}$ to synthesize methionine and requires functional BtuB for full growth on minimal medium in the presence of 0.1–1 nM vitamin B$_{12}$. The colony size of *E. coli* RK5016 (*btub metE*[38]) cells expressing wild-type BtuB or the crosslinked variants under the control of a native promoter (pAG1 (ref. 39)) grown on minimal medium agar plates supplemented with vitamin B$_{12}$ (0.1–100 nM) or methionine (5 μg ml$^{-1}$) were compared after 48 h (Table 1). Full growth was observed for wild-type and XL$_{loop}$ BtuB (Table 1) at all concentrations of vitamin B$_{12}$, while reduced growth was observed at all vitamin B$_{12}$ concentrations for BtuB XL$_{barrel}$. Full growth was observed for all BtuB variants in the presence of methionine (Supplementary Fig. 5). The phenotypes of these variants support the hypothesis that application of force (via TonB) unravels the mechanically weak plug subdomain from the N-terminus to create a channel between the plug and wall of the β-barrel.

To determine whether channel formation by force-induced remodelling of the plug via TonB is a feature general to all TBDTs, we repeated the force spectroscopy experiments on the ferric hydroxamate uptake (FhuA) receptor from *E. coli*, which transports ferrichrome (740.52 Da)—a significantly smaller cargo than vitamin B$_{12}$ (1355.38 Da). After purification and immobilization (see 'Methods' section), two rupture events (74 ± 11 and 105 ± 24 pN and 62 ± 4 and 85 ± 9 nm; Supplementary Fig. 6a and Supplementary Table 4) were detected reminiscent of those observed for BtuB. The change in contour length for FhuA, however, was found to be greater than for BtuB ($\Delta L_c = 25 \pm 1$ nm) (Supplementary Table 4 and Supplementary Fig. 6b),

corresponding to the removal of 63 residues (residues 20–83). Importantly, while remodelling once again results in the formation of a channel across the receptor (Supplementary Fig. 6c), its dimensions are slightly smaller (14 × 26 Å) reflecting the smaller size of its cargo.

**Destabilizing the mechanically strong plug subdomain.** The presence of different sized mechanically strong plug subdomains in BtuB and FhuA suggests that these TBDTs have evolved to limit their channel dimensions to allow passage of their substrate. To investigate the consequences of enlarging the channel to the dimensions of the β-barrel lumen (~35–40 Å diameter), the C-terminal plug subdomain of BtuB (residues 73–136) was destabilized by the introduction of three large hydrophobic deletion substitutions (I80A/L85A/L96A, designated as BtuB$_{3A}$). By contrast to wild-type BtuB and the other variants studied, the force–extension profiles for BtuB$_{3A}$–TonB$_{\Delta TMD}$ displayed three rupture events (Fig. 6a and Supplementary Table 5). The increase in contour length between the first and second rupture event ($\Delta L_{c1} = 19 \pm 4$ nm) was similar to that found for wild-type BtuB–TonB$_{\Delta TMD}$ ($\Delta L_c = 20 \pm 1$ nm) identifying this event as the unfolding of the mechanically weak subdomain. If the additional rupture event ($F_U = 94 \pm 9$ pN) observed for BtuB$_{3A}$ is due to the unfolding of the remaining 64 plug residues, an increase in contour length of ~25 nm would be expected, in good agreement with the measured value ($\Delta L_{c2} = 26 \pm 3$ nm). The difference in contour length between the first and last event (total $\Delta L_c = 45 \pm 6$ nm) is equivalent to the unfolding of 112 ± 13 residues, in good agreement with the number of residues in BtuB plug (114 (residues 22–136), Fig. 6b).

*In vitro*, the drastic weakening of the mechanically robust plug subdomain allows full plug unfolding before dissociation of the

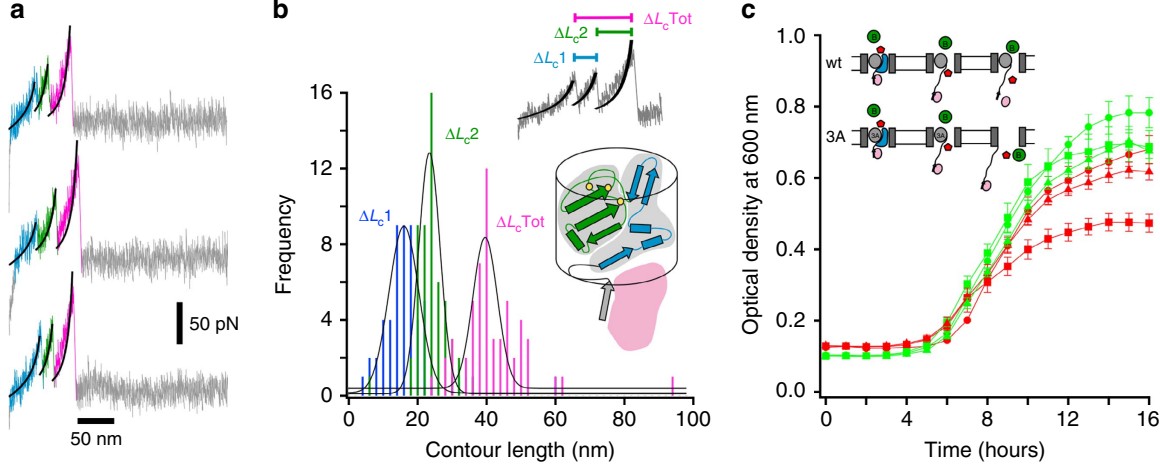

**Figure 6 | Destabilizing the mechanically strong subdomain leads to complete unfolding.** (**a**) Force–extension profiles for BtuB$_{3A}$–TonB$_{\Delta TMD}$ display triple rupture events. Each event is fitted to a WLC model (black). Data were obtained at a retraction velocity of 1,000 nm s$^{-1}$. (**b**) $\Delta L_c$ frequency histograms ($n = 55$) for the difference in $L_c$ between the first and second event ($\Delta L_{c1}$, blue histogram), the second and third event ($\Delta L_{c2}$, green histogram) and the first and third event ($\Delta L_{cTot}$, pink histogram) (see top inset). As the length of an amino acid is known (0.4 nm), these data can be used to calculate the number of amino acids unfolded at each event, allowing the unfolding events to be mapped onto the secondary structure schematic (bottom inset: residues unfolded at the first and second rupture events coloured blue and green, respectively. Yellow dots represent the location of each alanine mutation). (**c**) Bacitracin sensitivity assay. RK5016 cells transformed with BtuB or variants (circles: wild-type, squares: BtuB$_{3A}$ and triangles: BtuB$_{3A,L8P}$) under a native promoter were used to inoculate LB medium supplemented with selection antibiotics and 200 μg ml$^{-1}$ bacitracin in the absence (red data) or presence of 200 μg ml$^{-1}$ (green data) of Ton box pentapeptide in a 96-well format with shaking at 200 r.p.m. at 37 °C. The absorbance at 600 nm was monitored over 15 h. Data points are averages of three technical repeats with error bars showing s.d. Inset: the mechanically strong plug subdomain (grey ellipse) of wild-type BtuB does not unfold upon extension via TonB (pink ellipse). The channel formed by unfolding the mechanically weak domain (blue shaded region) thus allows passage of vitamin B$_{12}$ (red pentagon) but not bacitracin (green circle). By contrast, upon destabilizing the mechanically strong subdomain (BtuB$_{3A}$), extension via TonB results in the unfolding of both subdomains, leading to the formation of a channel that allows passage of both substrate and antibiotic.

TonB–BtuB complex. If this mechanical hierarchy is preserved *in vivo*, which would be necessary for any force-modulated gating mechanism, this would result in the formation of a transient $\sim 3.5 \times 3.5$ nm pore within the OM. Consequently, an antibiotic sensitivity screen was performed using bacitracin, a large (1.4 kDa) inhibitor of cell wall synthesis. *E. coli* are normally unaffected by bacitracin at concentrations less than $200 \, \mu g \, ml^{-1}$, as the OM provides a natural barrier to this large antibiotic[40]. If the integrity of the OM is compromised by addition of polymyxin B[40] or by the presence of a TBDT without a folded plug domain[41], sensitivity is increased. Accordingly, the growth of RK5016 cells expressing wild-type or variant BtuBs under the control of the native promoter was measured in the presence of $200 \, \mu g \, ml^{-1}$ bacitracin. Cells expressing $BtuB_{3A}$ exhibited reduced growth (typically reaching 50% of the maximal growth of cultures expressing wild-type BtuB, Fig. 6c) suggesting an increased accessibility of bacitracin to the periplasm. To rule out the possibility that the increase in bacitracin sensitivity was simply due to the inability of $BtuB_{3A}$ to form a stable natively folded plug domain, $200 \, \mu g \, ml^{-1}$ Ton box pentapeptide (ETVIV) was added to the growth medium to prevent active unfolding of the plug by TonB[42]. The addition of $200 \, \mu g \, ml^{-1}$ Ton box pentapeptide yielded a phenotype similar to wild-type BtuB (Fig. 6c). Introduction of the L8P substitution (known to disrupt formation of TonB–BtuB complex *in vitro*) within the Ton box of $BtuB_{3A}$ resulted in bacteria with significantly reduced sensitivity to bacitracin relative to $BtuB_{3A}$. These results demonstrate that $BtuB_{3A}$ is active and able to form a native-like luminal plug *in vivo*. More importantly, the differential sensitivity to bacitracin of wild-type and $BtuB_{3A}$ in the presence and absence of Ton box peptide accords with the *in vitro* AFM data above, suggesting that *in vivo*, TonB drives partial (wild-type BtuB) or complete unfolding ($BtuB_{3A}$) of the plug before complex dissociation.

## Discussion

The work presented here demonstrates that the TonB–$TB_{BtuB}$ interaction is sufficiently strong under extension to allow the mechanical unfolding of a defined region of the plug domain of BtuB to create a substrate channel through the receptor before its dissociation. This mechanism is similar to that originally proposed a decade ago[21], which was supported by an *in silico* study which showed that extension of $TB_{BtuB}$ by $\sim 20$ nm leads to the creation of a channel that would allow substrate passage[22]. The mechanisms of both inside-out energy transduction and the resultant vertical displacement of TonB by a distance similar to the width of the periplasm[43] are still unclear. Despite differences in their details, however, all the suggested models involve the formation of a force-transduction pathway, which apply mechanical force to the plug domain. In these models, the vertical displacement of $TB_{BtuB}$ that is required for channel opening may be a result of 'passive' processes (for example, variations in the width of the periplasmic space[44] and/or the differential diffusion of the IM and OM proteins[45]). Alternatively, active processes dependent on the proton motive force may play a role. This includes a conformational change of the TonB linker resulting in a shortening in the end-to-end length of some part of the linker region (this work), by a transition of residues 66–100 (ref. 46) from an extended polyproline type II to a shorter polyproline type I helix[14], or by the ExbBD-dependent[23,47] rotary motion of TonB itself.

The direct identification of two distinct subdomains in the luminal plugs of FhuA and BtuB rationalizes a wealth of previous observations from mutational and crosslinking studies. For example, the first 51 residues of the FepA plug domain were found to be more susceptible to labelling relative to residues 91–142 (ref. 48), while the introduction of crosslinks between the N- or C-terminal portions of the FhuA plug to the barrel were found to prevent or support ferrichrome transport, respectively[49,50]. A lack of plug domain cooperativity has also been observed *in vitro*: electron paramagnetic resonance experiments and trypsin cleavage have shown that the N-terminal regions of the BtuB and FhuA plug domains (respectively) unfold in the presence of 4 M urea[51,52] and that, for BtuB, the unfolded subdomain rapidly refolds upon removal of the denaturant[51]. Finally, previous AFM studies, which investigated the mechanical unfolding pathway of the barrel of FhuA (by adsorbing FhuA to the AFM tip in a non-specific manner in the absence of TonB) showed that the plug domain unfolded in two steps (residues 1–91, $F_U = \sim 50$ pN and residues 92–161, $F_U = 163 \pm 50$ pN at a retraction velocity of 2,200 nm s$^{-1}$ (ref. 53)).

Here we demonstrate that complete unfolding allows the free diffusion of large antibiotics, such as bacitracin, into the cell. This observation, taken with others[22,48,51,52], suggests that TBDTs have evolved to minimize the deleterious effect of compromising the outermost barrier of Gram-negative bacteria by allowing the remodelling of the fewest residues required for transit of their cargo. The mechanically recalcitrant C-terminal plug subdomain of BtuB identified here, together with subdomains identified in FecA and FhuA, have been shown by evolution-based statistical analysis to contain a network of conserved residues believed to be involved in the allosteric communication that drives the release of the Ton box into the periplasm upon ligand association[54]. We therefore propose that the plug domain is divided into two separate functional parts: a mechanically strong subdomain involved in this signalling and a mechanically weak subdomain that is unfolded to allow substrate transport. In this regard, it is interesting to note that the mechanically weak subdomains of BtuB (Fig. 5a) and FhuA (Supplementary Fig. 6d) both make direct contact with their ligands. As these subdomains are presumably pulled into the periplasmic space during ligand transport, it is possible that the substrate is also pulled through the receptor. Ligand-plug contacts are also present in the substrate-bound crystal structures of FecA[55], HasR[56], FpvA[57] and FptA[58], therefore, this could represent a universal feature of TBDTs. It should be noted, however, that vitamin B$_{12}$ was displaced by only 3Å during both generation and equilibration of the plug-open state in the MD simulations described above (Fig. 5b). This suggests that either diffusion of substrate through the channel occurs over a longer timescale than our simulations, or that other factors play a role in import.

Overall, this work shows that, in addition to the lipid bilayer, membrane proteins can be directly mechanically gated by a protein–protein interaction. More specifically, we have shown that the TonB–Ton box interaction has mechanical properties perfectly tuned for its function: it is strong enough to facilitate the passage of a specific ligand by partially unfolding sufficient residues in the N-terminal plug subdomain to allow transport yet is weaker than the remainder of the plug domain, allowing dissociation before the integrity of the protective OM is compromised.

## Methods

**Protein expression and purification.** $TonB_{ATMD}$ was cloned from the genome of *E. coli* BL21 (DE3) (Agilent Technologies). Primers designed to flank the target gene sequence contained a 5′ extension on the forward primer to encode for an NdeI restriction sequence, (His)$_6$ tag for purification and a cysteine residue for AFM surface attachment (forward primer: 5′-AATAATTAACATATGCATCAC CATCACCATCACGGCTGTCATCAGGTTATTGAACTACCTGC-3′). The 5′ end of the reverse primer contained a stop codon and a XhoI recognition sequence (reverse primer: 5′-GGAATTTGACTCGAGTTACTGAATTTCGGTGGTGC CGTTA-3′). The primers for $TonB_{CTD}$ were 5′-AATTTACAACATATGCA TCACCATCACCATCACGGATGTCCGGT TACCAGTGTGGCTTCA-3′

(forward) and 5′-GGAATTTGACTCGAGTTACTGAATTTCGGTGGTGCC
GTTA-3′ (reverse). The forward primer encoded for an NdeI restriction sequence,
(His)$_6$ tag for purification and a cysteine residue for AFM surface attachment. The
5′ end of the reverse primer contained a stop codon and a XhoI recognition
sequence. PCR (using vent DNA polymerase (NEB)) was used to amplify the gene,
and after digestion with NdeI and XhoI, the product was ligated into pET-23a(+)
digested with the same restriction endonucleases. After verification of the presence
and sequence of the cloned gene by DNA sequencing, pET23a-TonB constructs
were transformed into E. coli BL21 (DE3) cells and grown in LB medium
(100 μg ml$^{-1}$ ampicillin) at 37 °C with shaking to an optical density at 600 nm
(OD$_{600}$) of 0.7 before induction with 0.1% (w/v) IPTG. After 16 h of incubation at
37 °C, the cells were pelleted, resuspended in loading buffer (20 mM Tris-HCl pH
8.0, 300 mM NaCl, 20 mM imidazole, 2 mM DTT, 0.025% (w/v) sodium azide,
1 mM PMSF and 2 mM benzamide), disrupted by sonication and the insoluble
material pelleted by centrifugation. The supernatant was then used to resuspend
3 ml Ni Sepharose 6 Fast Flow resin (GE Healthcare) and TonB purified
(TonB$_{ΔTMD}$) by batch purification. After washing with loading buffer, the protein
was eluted (loading buffer plus 400 mM imidazole). The imidazole was then
removed by dialysis into 25 mM Tris pH 7.5, 128 mM NaCl, then further purified
(TonB$_{ΔTMD}$) by ion-exchange chromatography (Resource S column (6 ml, GE
healthcare)) using a 0–50% gradient of 1 M NaCl over four column volumes. For
TonB$_{CTD}$, size exclusion chromatography using a Superdex 75 26/60 gel filtration
column (GE Healthcare) equilibrated with 25 mM Tris-HCl pH 7.5, 128 mM NaCl
was used. The purified TonB constructs were finally dialysed into 25 mM Tris pH
7.5, 128 mM NaCl using a 5 kDa molecular weight cutoff membrane and stored at
−20 °C after flash freezing in liquid nitrogen. All purification steps for TonB
constructs were carried out at 4 °C in the presence of 1 mM PMSF, 2 mM
benzamidine.

**Labelling of TB$_{BtuB}$ peptide with Alexa Fluor 488.** The Ton box peptide from
BtuB (TB$_{BtuB}$, sequence: PDTLVVTANR**GSWSC** (non TB$_{BtuB}$ residues in bold)) was
fluorescently labelled with Alexa Fluor 488 C5 maleimide (ThermoFisher) by
creating a 1 mg ml$^{-1}$ solution of the peptide in 25 mM Tris-HCl pH 7.5, 128 mM
NaCl and adding 100 μl of 10 mg ml$^{-1}$ dye dissolved in DMSO in a drop-wise
manner. This was then stirred for 16 h at room temperature in the dark. The
reaction mixture was then loaded onto a pre-equilibrated Superdex peptide gel
filtration column 10/300 (GE Healthcare) pre-equilibrated with 25 mM Tris-HCl
pH 7.5, 128 mM NaCl. The chromatogram from the column contained two major
peaks, the first peak was analysed by liquid chromatography electrospray ionization
mass spectrometry, which confirmed the presence of the labelled peptide (predicted
Mw = 2303.4 Da, observed = 2303.8 Da). The labelling efficiency was estimated
using UV-absorbance spectroscopy.

**Microscale thermophoresis (MST).** MST was carried out using a Monolith
NT.115 series instrument (NanoTemper) using Alexa Fluor-488 as the fluorescent
probe and a blue LED filter (excitation 460–480 nm and emission 515–530 nm).
The fluorescently labelled TB$_{BtuB}$ peptide was adjusted to a final concentration of
500 nM (optimal fluorescence when diluted to 250 nM). TonB constructs were
concentrated to 1 mM using a 3 kDa cutoff membrane Vivaspin column 500 and
centrifugation at 15,000g. The concentration was determined by ultraviolet
absorbance spectroscopy (280 nm) and an extinction coefficient of 5,500 M$^{-1}$
cm$^{-1}$ (calculated using ExPASy ProtParam silico). A serial dilution over 16
concentrations was set-up for TonB in 25 mM Tris pH 7.5, 128 mM NaCl. This
dilution series was mixed with the 500 nM TB$_{BtuB}$-Alexa fluor 488 1:1 (v/v)
and incubated at room temperature for 5 min. The samples were then loaded into
capillaries (K002 Monolith™ NT.115 standard treated capillaries) and analysed
using a LED power set to 95% and MST power (IR-laser) set to 40%. The nor-
malized fluorescence (equation (1)) from T-jump and thermophoresis (F$_{norm}$):

$$F_{norm} = 1 + \left(\frac{\delta F}{\delta T} - S_T\right)\Delta T \tag{1}$$

(where $\delta F/\delta T$ is the fluorescence change due to temperature increase, $\Delta T$ is the
change in temperature and $S_T$ is the Soret coefficient) was then plot against the
log$_{10}$ of the protein concentration and the data were fitted using the Nanotemper
analysis software using the law of mass action:

$$FB = \frac{[AB]}{[B]} = \frac{[A] + [B] + K_D - \sqrt{([A] + [B] + K_D)^2 - 4[AB]}}{2[B]} \tag{2}$$

where FB is the fraction of bound B. $F_{norm}$ from the MST experiment yields the
FB[59].

**AFM cantilever derivatization.** Silicon nitride AFM probes (MLCT with reflective
gold, Bruker) and surfaces (1 cm$^2$ cut from a silicon nitride disc (Rockwood
electronic material)) were oxidized by submersion for 30 s in piranha solution
(0.5 M (>95% (v/v) H$_2$SO$_4$ and 30% (v/v) H$_2$O$_2$ mixed 3:1 v-v) followed by
washing with excess distilled and de-ionized H$_2$O and drying with a nitrogen flow.
The surfaces and AFM probes were then placed under a UV lamp (UVIlite,
UVItec) set to 254 nm for 30 min before being transferred into a desiccator along

with 80 μl of (3-aminopropyl)triethoxysilane (APTES) and 20 μl of N,-N-diiso-
propylethylamine held in separate 1.5 ml Eppendorf tube lids. The desiccator was
evacuated using a vacuum pump and left to incubate at room temperature for 2 h.
After the incubation, the APTES and N,-N-diisopropylethylamine solutions were
removed and the derivatized probes/surfaces allowed to cure in an N$_2$ atmosphere
for 48 h. The amino-silanized AFM probes and surfaces were then immersed in
1 ml chloroform containing 20 μl of 250 mM NHS-PEG$_{24}$-maleimide, 250 mM
NHS-PEG$_{24}$-methyl (1:9 ratio in DMSO) and left to incubate at room temperature
for 1 h.

Both AFM probes and surfaces were then washed with chloroform and dried
under a stream of nitrogen gas. Protein (1 mg ml$^{-1}$) and peptide (1 mg ml$^{-1}$) both
containing an engineered cysteine residue were deposited over the surfaces and
AFM probes and left to incubate in a covered container for 30 min at room
temperature. For TonB–TB$_{BtuB}$ dissociation, the peptide was attached to the AFM
probe and TonB to the derivatized surface. For TonB–TBDT dissociation, TonB
was attached to the AFM probe and the TBDT (inserted into a lipid bilayer)
adsorbed to mica (see below). Unreacted protein/oligopeptide were then washed
from the surface and AFM probe with buffer (25 mM Tris-HCl, pH 8.0 128 mM
NaCl).

**Force spectroscopy.** The derivatized AFM probe (Bruker MCLT (nominal spring
constant: 30 pN nm$^{-1}$)) was mounted on to a MFP-3D head (Asylum Research)
and calibrated using the thermal method[60]. Force maps of 20 μm$^2$ with 600
approach-retract cycles (starting 1 μm from the substrate surface) were taken
to ensure a good coverage of the surface. The approach velocity was kept
constant (1 μm s$^{-1}$) for all experiments while the retraction velocity was varied
(0.2–5 μm s$^{-1}$, see main text). All experiments were conducted in 25 mM Tris-HCl
pH 8.0, 128 mM NaCl at 25 °C. All experiments (that is, for each protein variant
and/or retraction velocity) were performed at least in triplicate (using a freshly
prepared cantilever for each repeat). After filtering and analysis (see below) using
IGOR pro 6.32A with an Asylum Research extension (MFP3DXopv30), single
Gaussian distributions were generated and fit to unbinding force and contour
length histograms in order to determine the most probable force and contour
length at rupture for each retraction velocity.

**Force curve analysis.** Generally less than 1 in 10 approach-retract cycles resulted
in a true unbinding event (average hit rates ranging from 5–10%). When a cognate
complex is extended via the PEG linkers, there is a large change in the force
response and hence a peak in the observed rate of change of the force response. To
ensure sufficient surface sampling, 20 μm$^2$ force maps with 600 approach-retract
cycles were taken, and the cantilever then repositioned to a new area. A typical data
set at a single retraction velocity contained 2–4 force maps, depending on the hit
rate. All detected events were then manually fitted to the WLC model[30]
(equation (3)), where the persistence length was fixed (0.4 nm)[61], the hard contact
was used to identify the zero distance and the retraction baseline was used to zero
the force.

Detected events for tip-sample separation values of less than 10 nm were in
general found to be consistent with non-specific tip-sample interactions and
displayed a linear force–distance profile. Force–extension profiles were binned for
analysis if: (1) the data fitted to the WLC model (single-molecule events should
display WLC like behaviour where the force–distance profile is not linear) and
(2) the contour length was greater than the length of the PEG$_{24}$ linkers used. For
the TonB$_{ΔTMD}$–TB$_{BtuB}$ experiments, fewer than 0.1% of force curves showed more
than one event. This indicates that there are no detectable unfolding events for
either protein domain before dissociation of the bound complex.

Loading rates were calculated by fitting a WLC model to the rising edge of each
unbinding profile when plotted as force versus tip-sample separation. The
instantaneous gradient of this fit at rupture (WLC$_{slope}$) was calculated by inserting
the derived contour length and extension at rupture into a differentiated form of
the same equation (equation (4)). The loading rate at rupture was then obtained by
multiplying this value by the retraction velocity.

$$f(x) = \frac{k_B T}{p}\left(\frac{1}{4}\left(1 - \frac{x}{L_c}\right)^{-2} - \frac{1}{4} + \frac{x}{L_c}\right) \tag{3}$$

$$WLC_{slope} = \frac{k_B T}{p}\left(\frac{1}{2L_c\left(1 - \frac{x}{L_c}\right)^3}\right) + 1/L_c \tag{4}$$

where $f$ is the force acting on the WLC, $p$ is the persistence length, $L_c$ is the contour
length, $x$ is the extension, $k_B$ is Boltzmann's constant and $T$ is the temperature.

**Estimation of the contour length.** The contour length estimation of the
unstructured linker domain of TonB$_{ΔTMD}$ was made on the basis of the known
length of the PEG$_{24}$ linkers and the known length of TonB$_{CTD}$ in complex to
TB$_{BtuB}$ (PDB: 2GSK), with any additional length observed due to the proline-rich
(PR) domain (Supplementary Fig. 2b). The length of a stretched amino acid is
0.4 nm; therefore, if the PR domain (118 amino acids) behaved as an unstructured
polypeptide chain, the expected length would be 47.2 nm giving a total contour

length at detachment of ~72 nm. For the TonB$_{CTD}$ construct (lacking the PR domain), the expected length would be ~24.5 nm.

### Purification of BtuB and FhuA.

pNGH015 (a pBAD expression plasmid encoding wild-type BtuB, a gift from Professor C. Kleanthous, University of Oxford) or pBAD-FhuA (see below) was transformed into *E. coli* TNE012 (K12 *tsx⁻ ompA⁻ ompB⁻*) cells (also provided by C. Kleanthous) and a single colony then used to inoculate LB medium (100 µg ml⁻¹ ampicillin). The cells were cultured at 37 °C with shaking at 200 r.p.m. to an OD$_{600}$ = 0.7 and then induced with 0.15% (w/v) L-(+)-arabinose. After induction, the temperature was decreased to 15 °C and incubated for a further 16 h. The cells were collected by centrifugation (9500g at 4 °C), resuspended in 50 ml 10 g⁻¹ buffer (10 mM Tris-HCl, pH 8.0, 0.25% (w/v) lithium diiodosalicylic acid), disrupted by sonication (on 3 s, off 7 s for total process time of 4 min) and the insoluble material removed by centrifugation (10,000g at 4 °C). The supernatant (containing membranes) was subjected to ultra-centrifugation (158,000g) to collect the total membrane fraction. The membrane pellet was resuspended in 50 ml 10 mM Tris-HCl, pH 8.0, 0.25% (w/v) LIS, 2% (v/v) Triton X-100, homogenized by pipetting and then ultracentrifuged at 158,000g for 1 h at 4 °C. The supernatant was discarded and this step was repeated. The pellet was then resuspend in 56 ml BtuB purification buffer (10 mM Tris-HCl, pH 8.0), homogenized and ultracentrifuged at 158,000g for 1 h, 4 °C and the supernatant was discarded. The pellet was resuspend in 56 ml 10 mM Tris-HCl, pH 8.0, 1% (w/v) n-octyl-β-D-glucopyranoside (β-OG), 5 mM EDTA, homogenized thoroughly using a hand homogenizer and ultracentrifuged at 158,000g for 1 h, 4 °C. The supernatant containing the extracted OM proteins was then loaded onto a 5 ml DEAE-Sepharose column (GE Healthcare) pre-equilibrated in 90% buffer A (50 mM Tris-HCl, pH 7.5, 5 mM EDTA, 0.54% (w/v) β-OG), and 10% buffer B (50 mM Tris-HCl, pH 7.5, 5 mM EDTA, 1 M LiCl, 0.54% (w/v) β-OG) at room temperature using a AKTA prime plus (GE Healthcare). The OM proteins bound to the column were initially washed with 40 ml 90% buffer A, 10% buffer B. A gradient over 65 ml of 10–50% buffer B was then run to remove impurities. Finally, 30 ml of 100% buffer B was run through the column, which causes the target protein to elute from the column. Shortly after elution from the column, pure BtuB (and its variants) forms a white precipitate, which was dissolved into 1% (w/v) β-OG. By contrast to BtuB, FhuA eluted at 22% 1 M LiCl with impurities. A white precipitate formed after elution and was found to contain FhuA by SDS–PAGE, which was isolated and then solubilized in 1% (v/v) β-OG. A Superdex 75 10/300 GL column was used to further purify FhuA in 25 mM Tris-HCl pH 8.0, 1% (w/v) β-OG. The major peak eluted from this column was identified as FhuA by SDS–PAGE and circular dichroism spectroscopy and fluorescence spectroscopy was used to confirm the correct folding.

### Force spectroscopy of TBDTs in lipid membranes.

Using the method described in ref. 53, TonB-dependent receptors were inserted into a lipid bilayer at a high density by solubilizing the precipitated receptor (0.1 mg) in 200 µl 25 mM Tris-HCl, pH 7.5, 128 mM NaCl, 1% (w/v) β-OG and dialysing in the presence of lipids (0.2 mg) (*E. coli* polar lipid extract (Avanti)) dried from a chloroform stock (10 mg ml⁻¹) using a nitrogen flow. The ligand for the receptor (cyanocobalamin (Sigma) and ferrichrome (EMC Microcollections) for BtuB and FhuA, respectively) was added to the buffer at a 100-fold molar excess of the receptor. The dried lipids were then mixed with the protein/ligand detergent solution and lipid was solubilized by pipetting. The components were then placed into a 200 µl 12–14 kDa MWCO dialysis bag (D-tube dialyser mini, Merck Millipore) and dialysed against 300 ml of detergent-free buffer (10 mM Tris-HCl, pH 7.5, 300 mM KCl, 400 µM CaCl$_2$, 0.01% (w/v) NaN$_3$) for 7 days at room temperature with daily buffer changes. The sodium azide was omitted for the last dialysis stage. The proteoliposomes were stored at 4 °C. Vesicle formation was confirmed using dynamic light scattering and the presence of TBDT by SDS–PAGE.

To immobilize TBDT-containing liposomes for study by AFM, the vesicles were resuspended by pipetting and then diluted to a final lipid concentration of 0.1 mg ml⁻¹. An aliquot of 200 µl was pooled over a 1 cm diameter disc of freshly cleaved mica and incubated at room temperature for 30 min. The surface was then thoroughly rinsed with 25 mM Tris-HCl pH 8.0, 128 mM NaCl. Force spectroscopy experiments were carried out as described above, using a cantilever derivatized with TonB$_{ΔTMD}$-NHS-PEG$_{24}$-maleimide, a contact force of 90–120 pN, a dwell of 1 s and a retraction velocity of 1,000 nms⁻¹ over 600 nm unless stated otherwise.

### Generation of BtuB variants and cloning of FhuA.

Site-directed mutagenesis was performed using the Q5 Site-Directed Mutagenesis kit (NEB). Primers were designed and purchased from Eurofins MWG Operon (Supplementary Table 6). Sequencing was used to confirm the success of the mutagenesis. Wild-type FhuA was cloned from the genome of *E. coli* JM109 cells (Agilent Technologies) using PCR (Q5 polymerase (NEB)) using the following primers: 5′-CATTAATCACCATGG CGCGTTCCAAAAACTGC-3′ (forward) and 5′-TTGAATATTCTCGAGTTAGAA ACGGAAGGTTGCGGTTG-3′ (reverse). The amplified product was then inserted into the pBAD vector used for BtuB expression by digesting with NcoI and XhoI restriction enzymes. After ligation (T4 ligase (NEB)), the pBAD-FhuA vector was transformed into TNE012 (K12 *tsx⁻ ompA⁻ ompB⁻*) cells and protein expression was induced with arabinose using the same protocol as described for BtuB.

### Molecular dynamics simulations.

MD simulations were performed using NAMD (v2.10)[62] and the CHARMM27 force field[63]. System preparation was performed using VMD (v1.9.2)[64]. 1-palmitoyl-2-oleoyl-sn-glycero-3-phosphocholine (POPC) was used to model the lipid bilayer, the parameters for vitamin B$_{12}$ were obtained from ref. 65, whereas the TIP3P force field was used for water[66]. The X-ray crystal structure of BtuB in complex with TonB and vitamin B$_{12}$ (PDB 2GSK), including calcium ions and crystal waters, was inserted into the homogenous POPC lipid bilayer placed in the XY plane. The dimensions of the simulation box were 100 × 100 × 300 Å (Supplementary Fig. 4a). A total of 100,211 molecules of TIP3P water, 149 Na⁺ and 141 Ca²⁺ were then used to fill the simulation box. The total system comprised 342,843 atoms. The simulations were conducted using periodic boundary conditions. The bonds between hydrogen and heavy atoms were constrained with SHAKE[67]. The r-RESPA multiple time step method[68] was employed with 2 fs for bonded potentials, 2 fs for short-range non-bonded potentials and 4 fs for long-range non-bonded potentials[62]. Long-range electrostatic interactions were treated with the particle-mesh-Ewald method[69]. The distance cutoff for non-bonded interactions was set to 10 Å, and a switching function was applied to smooth interactions between 9 and 10 Å. All simulations were conducted in the NPT ensemble. The temperature was set to 300 K and regulated via a Langevin thermostat[70]; the pressure was set to 1 atm and regulated via an isotropic Langevin piston manostat[71].

A multistage equilibration was used to relax the system. Initially all of the atoms except water and ions were fixed. Initially, 2,000 steps of conjugate gradient geometry optimization were conducted. After energy minimization, the system was simulated for 1 ns to equilibrate the solvent around the protein and membrane. Next, the membrane was relaxed around the protein by removing the restraints on the fatty acid tails for 1 ns, and keeping the protein and vitamin B$_{12}$ atoms restrained to their initial positions by applying an harmonic potential on the heavy atoms using a force constant of 1 kcal (mol Å²)⁻¹. The protein and ligand was then gradually released over the next 3 ns by decreasing the force constant to 0.5 kcal (mol Å²)⁻¹. Finally, all restraints were removed and the system was equilibrated for 10 ns (Supplementary Fig. 4b). The final structure from the unrestrained MD trajectory was used as the starting point in the pulling simulations (Supplementary Fig. 4a). The pulling was performed by using the steered molecular simulation (SMD) module as implemented in NAMD[62]. The SMD simulation was performed by restraining the phosphate head groups of the upper bilayer leaflet to its initial coordinates (to prevent the membrane being dragged) and extending a harmonic spring attached to either the Ton box of BtuB (Pro5 Cα) or the centre of mass of TonB (all Cα residues of TonB). The spring constant was set to 0.5 kcal (mol Å²)⁻¹, while the pulling velocity was set to 2.5 Å ns⁻¹ along the z-direction.

### BtuB methionine phenotype assay.

*E. coli* RK5016 cells[72] (MC4100, *metE70, argH, btuB, recA*, a gift from Professor C. Kleanthous, University of Oxford) freshly transformed with pAG1 encoding wild-type *btuB* or its variants were streaked onto agar plates containing Davis and Mingioli minimal medium supplemented with 5 µg ml⁻¹ arginine, 1 µg ml⁻¹ thiamine, 100 µg ml⁻¹ ampicillin, 25 µg ml⁻¹ streptomycin and various concentrations of vitamin B$_{12}$ (0.1, 1, 10, 100 nM) or 5 µg ml⁻¹ methionine. These were incubated at 37 °C for 48 h before analysis of colony size. Full confluent growth (thus functional BtuB) is indicated by large colonies (diameter 1.5–2 mm) on the vitamin B$_{12}$-supplemented plates that are comparable with the methionine-supplemented control (Supplementary Fig. 5).

### Bacitracin sensitivity assay.

Single colonies of RK5016 cells transformed with pAG1 encoding wild-type *btuB* or its variants were used to inoculate 96-well plate wells containing 200 µl LB supplemented with ampicillin (100 µg ml⁻¹), strepto-mycin (25 µg ml⁻¹), 200 µg ml⁻¹ bacitracin and for a control where TonB activity is prevented, 200 µg ml⁻¹ Ton box pentapeptide (ETVIV). The plates (Corning Costar assay plate, 96-well, black with clear flat bottom, non-treated, 1 cm path length) were incubated at 37 °C with 200 r.p.m. shaking for 24 h and the absorbance at 600 nm measured every 4 min (CLARIOstar high-performance monochromator multimode microplate reader (BMG LABTECH)).

### Data availability.

The authors declare that the data supporting the findings of this study are available within the article and its Supplementary Information files and from the corresponding author upon reasonable request. The following PDB structures with accession codes 2GSK, 1NQE and 1XX3 were used in this work.

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

## Acknowledgements

We thank Dr Nicholas Housden and Professor Colin Kleanthous (University of Oxford) for providing plasmids pNGH015, pAG1 and *E. coli* strains RK5016 and TNE012 together with advice on purifying BtuB. Thanks to Dr James Gumbart (Georgia Tech) for providing the vitamin B$_{12}$ parameters in a usable format. We also thank Professor Sheena Radford and members of the Brockwell/Radford groups for useful discussions. This work was supported by the Wellcome Trust Grant 099753/Z/12/Z (to S.J.H.).

## Author contributions

S.J.H. designed and performed all experiments and analysis, R.E.M.C. created and purified the BtuB$_{3A}$ mutant, L.B. set up the MD simulation system, E.P. coordinated the MD simulation experiments, D.J.B. coordinated the entire study, designed experiments and wrote the manuscript with S.J.H.

## Additional information

**Competing interests:** The authors declare no competing financial interests.

**Publisher's note**: 

