## [Peer Review File · Nature Communications]

Reviewer #1 (Remarks to the Author)

Hickman et al. investigate the mechanical forces involved in the gating of TonB Dependent Transporters by the TonB protein using Atomic Force Microscopy. By linking C-terminal domains of the TonB protein to the tip of the AFM probe, they measure the force required to partially unfold the plug domain of an immobilized TBDT, and to break the interaction between the TonB domain and the TBDT TonB box. The correlated distances are in good agreement with the reported length of the TonB periplasmic domain, up to 20nm, which is supposed to span the periplasmic space.

In this paper, the authors are basically testing the TonB pulling hypothesis that was defined by molecular dynamics experiments for BtuB in 2009 and for TbpA in 2012. Unfortunately these experiments are not discussed in the manuscript, meaning that insufficient attention to background literature was given. According to the model, the TonB C-terminal domain binds the TBDT TonB box, and the TonB complex exerts some force from the inner membrane to pull down the TonB Cterm domain linked to the TonB box, to eventually unfold the TBDT plug domain so as to open a channel in the TBDT. The authors find that the interaction between the TonB box and the C-terminal TonB domain is strong enough to sustain the unfolding of about 60 residues of the plug domain before the interaction with the TonB box breaks down. While not proving the pulling hypothesis, the experimental data are in good agreement with the expected values. Furthermore different mutants are tested with the experimental setup, as well as in vivo, which seem to agree well with the opening of a channel through the TBDT.

While this referee is not a specialist of the Atomic Force Spectroscopy technique, the data presented seem consistent and strongly support the pulling theory as viable. Importantly it shows that the interaction between the TonB C-terminal domain and the TonB box is strong and can resist the pulling and unfolding of the TBDT plug subdomain.

Specific Remarks:

[1] Page 3: many bacteria in addition to UPEC strains have TBDTs that function as virulence factors.

[2] Please cite the colicin review by Casales et al. after the last sentence in paragraph 1 on page 3. The citation provided is extremely out of date.

[3] In figure 1B the authors describe the interaction between the N-terminal plug domain (blue) and the C-terminal domain of TonB (pink). However, the TonB domain is labeled N as well. This is reiterated on page 4, 2nd paragraph. The authors need to clarify the setup and modify figure 1B.

[4] In Fig2A, the cartoon is said to represent the schematic setup of the AFM experiment, showing that the TonB Δ TMD is cross-linked to the silicon surface. However in all the experiments the TonB Δ TMD is cross-linked to the AFM probe. Is it a mistake in the figure? Or is this the actual used setup for this particular experiment? The methods section is not clear on whether it is the TonB Δ TMD or peptide that is linked to the AFM probe. In the supplementary Fig2 panel C, it is also shown that the TonB Δ TMD is linked to the surface and not on the probe.

[5] Page 7, 2nd paragraph, 1st sentence: "individual the" should be "the individual"

[6] Page 8: discuss the available in silico pulling experiments either here or in the discussion, comparing and contrasting results and predictions.

[7] Page 8, 2nd paragraph, last sentence: it is said that a minority of observations displayed a single rupture event, which is indicated by 31%. However, the value of 31% represents roughly a third of the events, which is not really a minority. Could the author elaborate on this?

[8] The authors used two engineered mutants of BtuB with cysteine (XL-loop and XL-barrel).. The data presented assume that these cysteines are forming disulfide crosslinks between domains of the BtuB plug and barrel, however there is no experimental data showing that these crosslinks are formed. Please also describe these mutants in structural terms, for example, 'the L23C/S374C XL-barrel mutation tethers barrel strand x to plug strand or helix or turn y'. Similarly describe the XL-loop mutation. Also include residues mutated in Table 1.

[9] Concerning the destabilizing mutant made by removing 3 large hydrophobic residues, The authors should show that this protein is expressed in cells at similar levels to WT and also that it remains heat-modifiable in isolated membranes. If folding or OM insertion is an issue with this mutant, the force extension profiles will be misleading.

[10] Supplementary Fig2, in Detailed BtuB purification section: the centrifugation steps are indicated with rpm without mentioning the rotor. It would be better to indicate the g forces.

[11] Although not mentioned in this manuscript, the authors should really discuss the contribution of the proline-rich periplasmic domain of TonB to force propagation. What is known about this type of structure generating rigidity or force in general? What role might it play in the TBDT activation?

Reviewer #2 (Remarks to the Author)

In the work presented Hickman et al. use AFM-based single-molecule force spectroscopy (SMFS) to elucidate the mechanical gating mechanism of bacterial TonB dependent outer membrane transporters, exemplified with the Vitamin B12 receptor BtuB of Escherichia coli. In particular they probed the interaction between the C-terminal receptor-binding domain of TonB, and the N-terminal globular plug domain of BtuB. The experiments yield evidence for a gating mechanism, which involves TonB to exert mechanical force on the plug domain resulting displacement of the plug domain from the barrel lumen. Such a process has been proposed several times over the last decades but has never been experimentally proven. The results presented by Hickman et al. provide biophysical basis for this mechanism by demonstrating i) that the interaction between TonB and the TonBox is sufficiently strong to allow partial mechanical displacement of the BtuB plug domain and ii) that the plug domain comprises a mechanically weak subdomain which unfolds under force and a mechanically stable domain which resists unfolding. While these results still leave some crucial aspects of the mechanism unanswered – for example how ligand transport is linked to mechanical gating and if the periplasmic lumen is spacious enough to support the mechanical unfolding over pulling distance of 20 to 25 nm – they considerably contribute towards a better understanding of this intricate protein machinery.

The majority of the experiments performed by the authors is based on force spectroscopy, which seems to be an adequate approach to study a force-based gating mechanism. By tethering on component to the AFM cantilever and immobilizing the other on the support the authors probed the behavior of multiple interaction partners under force: First, the authors showed that the TonB-TonBox interaction can withstand forces of ~100 pN, by tethering TonB and variations of the TonBox peptide. They then observed that when applied to reconstituted BtuB, the plug domain of BtuB partially unfolds before the interaction between TonB and the plug domain ruptures. Partial unfolding of the plug domain was then confirmed by introducing two different disulfide-bridge forming mutations in BtuB, which would prevent or interfere with partial unfolding of the plug domain. Partial unfolding of the plug domain was further verified by repeating the experiments with reconstituted FhuA. All force spectroscopy experiments were performed in agreement with current standards in the field with regard to experimental design, realization and analysis. Some minor changes are suggested below in detail.

In addition to the SMFS experiments, the authors tried to rationalize their findings using molecular dynamics (MD) simulations as well as to investigate their applicability in vivo. It appears as if the MD simulations and in vivo experiments cannot fully keep up with the quality of the SMFS

measurements. Especially the MD simulations are poorly integrated in the manuscript and do not yield much novel insight (Gumbart et. al presented similar simulations in much more detail in 2007). To a lesser extent this also applies to the in vivo assays, which lack experimental details such as the number of replicates performed. Suggested changes are addressed below in detail.

In all figures showing SMFS experiments the number of analyzed events should be given either directly in the figure or in the figure legend.

In figure 6b only the outline of the frequency histograms is shown. I suggest the authors should show the histograms in a bar-representation (as they do in the preceding figures) and include the Gaussian functions fitting the distributions.

For readers who are not familiar with MD simulations the RMDS plots in Figure 5 C and D appear rather inconclusive. Also the figure legend lacks information in this context. As these plots show that the analyzed fraction of the protein maintains its unperturbed folded conformation this should be described accordingly. I would suggest also introducing the meaning of RMSD as a measure of distance between 2 structures (in this case perturbed and unperturbed).

In Figure 5 A and B it appears that in the MD simulations the ligand remains stably bound to the extracellular cavity of BtuB, even after forced unfolding of the weak plug subdomain (a phenomenon we can confirm from similar simulations with FhuA). This could indicate that the mechanism requires more than just the mechanical opening of a transient channel and should be mentioned and discussed in the manuscript.

Table 1 shows the result of an in vivo growth assay. How often was this assay performed? The number of replicates should be included in the figure legend.

In the same assay the authors observed partial growth of colonies expressing the XL barrel mutation. Reduced growth indeed correlates with ~74% of the BtuB molecules showing only one rupture event due to formation of a disulfide bridge (Figure 4). However, for a fully locked BtuB mutation one would expect to observe a null-growth phenotype. This could be achieved by repeating the growth assay in the presence of oxidizing agents such as CuSO₄.

Figure 6 C shows growth curves of bacteria. How often was this assay performed? Pls provide the statistics. Do the traces represent average curves of multiple experiments? As such growth assays sometimes show large variability between individual cultures of the same strain I suggest also including error bars/confidence intervals and statistical tests. Furthermore the number of replicates should be included in the figure legend.

The supplementary list of BtuB mutagenesis primers includes an entry "BtuB Y109C". I cannot find such a mutation anywhere in the manuscript and suggest removal of the entry.

The histogram shown in Fig 3D should be fitted with a multi Gaussian instead with three separate Gaussian functions. Particularly the Gaussians fitting the L1 and L2 distributions show a large overlap and thus considerable data sets are fitted (accounted) twice.

Each histogram shown in Fig. 4 should be fitted with a multi Gaussian. As its done currently the data is not well fitted and the validity of the fits is questionable.

The authors interpret the rupture forces of 84-132 pN to be 'remarkably strong' compared to the low affinity. Such comparison must be done with great caution - as the pathway taken to mechanically rupture the bond may not be the same as measured in affinity measurements. The authors should revise their statement accordingly.

The authors write: "The lipid vesicles containing BtuB were then rolled onto a freshly cleaved mica surface and TonB Δ TMD immobilised to the AFM probe (Figure 1C)." It is not clear what 'rolled'

means. Do the authors want to say that they adsorbed the lipid vesicles onto the mica? I have so far not been aware that vesicles can be rolled onto mica. In addition Fig. 1C shows a lipid bilayer on mica not a vesicle. Please revise to remove confusion.

The authors write: "We have demonstrated above that TonB Δ TMD:TBBtuB is able to resist relatively high forces before dissociating." High is confusing here (and at some other passages of the text). Without a comparison high is meaningless. Forces around 100 pN may be rather small for others.

In the discussion the authors should clearly elaborate on which crucial aspects of the mechanism remains unanswered by their experimental setup and how the remaining questions may be addressed in the future. For example it is not clear how ligand transport can be linked to a mechanical pulling distance of 20 to 25 nm, particularly because it is not known how the periplasmic lumen can undergo such movements. Thus it may rather be expected that not increasing the pulling distance per se will lead to the unfolding of the plug domain but rather a mechanical stress applied – such as in principle could be applied by force clamp measurements.

Reviewer #3 (Remarks to the Author)

The paper by Hickman et al. reports on a large set of experiments aimed at understanding TonB-dependent transport. The authors primarily use AFM to determine the force response of a complex between TonB and BtuB (as well as FhuA) under different conditions and mutations. The results confirm existing theories about a force-driven gating of these outer-membrane transporters, as well as provide a hypothesis that each is tuned to the size of its substrate.

This is an impressive piece of work that has been long overdue in this field. The number of experiments exploring different possibilities make a strong case for the authors' conclusions. However, I would like to make some suggestions for improvement.

Main issues

1) I find the presentation confusing at times. It may be that there is just a lot of data to present, but I found myself constantly checking what a given name meant or what a particular length should be. It might be a given that one has to read such a paper very carefully, but anything the authors' can do to further improve clarity would be appreciated.

2) Page 7, paragraph beginning with "Closer examination...": This entire paragraph is confusing. First, the latter value is 74 nm, yet it is said to be consistent with 47.2 nm; I assume the authors mean the former value? Also, it is not clear to me why TonB must also have a conformation 20.5 nm in length based on the former value (or the latter?). Finally, why is 24.5 nm expected?

Finally, it seems like this paragraph comes too soon in the text; expected and average lengths don't come up until a few pages later (Fig. 3).

3) Throughout: combined units are unclear. The meaning of "nms⁻¹" requires context to deduce. The authors should insert a dot between nm and s⁻¹ (for example) every time a combined unit appears.

4) I would like the authors to speculate on how pulling over such large distances may be possible. The periplasm is only ~25 nm wide! Perhaps this fits with the fact that deltaLc usually ends up around 20-25 nm? But where is all the slack going? Does the TonB CTD end up touching the inner-

membrane complex by the end?

5) Can the authors address the TonB “dimer vs. monomer” debate with their results?

Minor issues

1) In the introduction, the authors should cite the newly released structure of ExbB and ExbD:
<http://www.nature.com/nature/journal/vaop/ncurrent/full/nature19757.html>

2) Page 7: “Closer examination of individual the” - “the” should come before “individual”.

3) Figure 3 caption: “dissoicates” is misspelled.

4) Page 10: “The Lc value is in accord...while the former value is close...” The authors should make clear that the “former value” is actually F_U.

5) Page 10: “Most importantly, the ΔL_c -frequency histogram (Figure 4C grey histogram) yielded a significantly shorter modal value relative to wild-type (14 ± 3 nm and 20 ± 1 nm), confirming unfolding in the N-terminal portion of the plug.”

I wouldn't say this is “significantly shorter”. Maybe just say it's 6 nm less?

6) Page 11: What is a “stringent concentration”? Maybe just say sufficient?

Reviewer 1

[1] Page 3: many bacteria in addition to UPEC strains have TBDTs that function as virulence factors.

We apologise for this error. The sentence now reads: “Their importance to cell viability results in TBDTs being **virulence factors in pathogenic bacteria**”. In addition we also include a second reference (reference 7).

[2] Please cite the colicin review by Casales et al. after the last sentence in paragraph 1 on page 3. The citation provided is extremely out of date.

The Kadner reference has been replaced by Cascales, E. *et al.*(2007) as requested.

[3] In figure 1B the authors describe the interaction between the N-terminal plug domain (blue) and the C-terminal domain of TonB (pink). However, the TonB domain is labeled N as well. This is reiterated on page 4, 2nd paragraph. The authors need to clarify the setup and modify figure 1B.

We thank the referee for highlighting this potentially confusing nomenclature. The pink protein in Figure 1B is indeed the C-terminal domain of TonB. The pink “N” thus denotes the N-terminus of the C-terminal domain. To clarify this, the legend to figure 1B now reads “Detail of the TonB:BtuB interaction showing the parallel orientation of the β -strand augmentation interaction of the Ton box of BtuB (blue) with TonB_{CTD} (pink). **Note: for TonB_{CTD}, N designates the start of the C-terminal domain.**”

[4] In Fig2A, the cartoon is said to represent the schematic setup of the AFM experiment, showing that the TonB Δ TMD is cross-linked to the silicon surface. However in all the experiments the TonB Δ TMD is cross-linked to the AFM probe. Is it a mistake in the figure? Or is this the actual used setup for this particular experiment? The methods section is not clear on whether it is the TonB Δ TMD or peptide that is linked to the AFM probe. In the supplementary Fig2 panel C, it is also shown that the TonB Δ TMD is linked to the surface and not on the probe.

The figures are correct. When using Ton box peptide (TB_{BtuB}), TonB was immobilised to the silicon surface (as shown in Figure 2). When using BtuB reconstituted into liposomes, TonB was immobilised onto the tip (as shown in Figure 1). To make this point clearer, the following sentence has been added to the “AFM cantilever derivitisation” section of the Methods: “**For TonB:TB_{BtuB} dissociation, the peptide was attached to the AFM probe and TonB to the derivitised surface. For TonB:TBDT dissociation, TonB was attached to the AFM probe and the TBDT (inserted into a lipid bilayer) adsorbed to mica (see below).**”

[5] Page 7, 2nd paragraph, 1st sentence: “individual the” should be “the individual”

Corrected.

[6] Page 8: discuss the available in silico pulling experiments either here or in the discussion, comparing and contrasting results and predictions.

As recognised by reviewer 2 our *pulling* simulations yielded consistent results to that described by Gumbart *et al.* and so we did not focus extensively on these details in the main text (they were, however, discussed in the Supplementary Information). The aim of MD simulations was to verify the experimental observation that approximately half of the plug domain appeared to resist mechanical unfolding – a surprising result given that the plug appears to possess a single hydrophobic core. The MD results replicated our experimental observations. This is described in the text and in Figure 5 b-d. To highlight the previous MD simulations we added a reference to Gumbart *et al.*'s simulations in the Introduction and now

explicitly state the similarity of the MD results (and another TBDT from *Neisseria*) in the Results section: “To assess the consequence of unfolding half of the plug and to visualise the process, we performed molecular dynamics (MD) simulations of the system. **These simulations were similar to that undertaken by Gumbart and colleagues in 2007 apart from the presence of bound vitamin B₁₂ in this study (Supplementary Note 4).** After the system was equilibrated (Supplementary Fig. 4), a force ramp was applied by retracting a harmonic spring at constant velocity (also sometimes called steered MD) until the Ton box was extended 20 nm away from its original position (Figure 5B, left). **Similar to the previous MD study on BtuB:TonB²² (and also observed for the *Neisseria* TbpA:TonB complex³⁷),** the plug domain unfolded directly downstream of the Ton box creating a continuous channel through the receptor (Figure 5B, right).”

We also discuss the key difference between the simulations in the discussion: “**It should be noted, however, that vitamin B₁₂ was displaced by only 3 Å during both generation and equilibration of the plug-open state in the MD simulations described above (Figure 5B).** This suggests that either diffusion of substrate through the channel occurs over a longer timescale than our simulations, or that other factors play a role in import.”

[7] Page 8, 2nd paragraph, last sentence: it is said that a minority of observations displayed a single rupture event, which is indicated by 31%. However, the value of 31% represents roughly a third of the events, which is not really a minority. Could the author elaborate on this?

We apologise for using such a qualitative expression. The ratio of single:double events simply gives a qualitative idea of the relative barrier heights and their position relative to the un-extended state. This section now reads: “**Approximately a third of the profiles (31 %) displayed a single rupture event ($F_U = 97 \pm 8$ pN, $L_c = 51 \pm 3$ nm) similar in force to that observed for the minimal TonB:TB_{BtuB} interaction at the same loading rate ($F_U = 113 \pm 7$ pN).** The majority, however, displayed two rupture events (Figure 3A) at F_U and L_c values of 61 ± 4 and 91 ± 23 pN and 58 ± 3 and 77 ± 7 nm (Supplementary Table 1). **These more frequent events indicate that the complex must undergo some partial unfolding before dissociation and that unfolding is more likely than dissociation of the complex at this loading rate.**”

[8] The authors used two engineered mutants of BtuB with cysteine (XL-loop and XL-barrel). The data presented assume that these cysteines are forming disulfide crosslinks between domains of the BtuB plug and barrel, however there is no experimental data showing that these crosslinks are formed.

The AFM experiments show clearly that the cross-links are formed as the mechanical phenotypes of these variants are different in the absence or presence of a reductant (see Figure 4).

Please also describe these mutants in structural terms, for example, ‘the L23C/S374C XL-barrel mutation tethers barrel strand x to plug strand or helix or turn y’. Similarly describe the XL-loop mutation. Also include residues mutated in Table 1.

We have now described the position of these residues as requested. The position of these residues are also shown schematically in the relevant figures. The text now reads: “**Firstly, a variant designated XL_{barrel} (Figure 4A schematic) was designed to prevent any unfolding by linking the N-terminus of the plug domain (residue 23 located between the Ton box and the first β -strand of the plug domain) and residue 374 in strand 12 of the barrel domain (BtuB L23C/S374C, yellow filled circles joined by red bar, Figure 4A, left)**” and “**To localise the unfolding event further, a V29C/V45C variant (XL_{loop}) was designed to cross-link strand 1**

(V29C) and helix 2 (V45C) of the plug domain (yellow filled circles joined by red bar, Figure 4A, right). Upon disulphide bond formation this creates a covalent 15 amino acid loop within the region predicted to unfold....”

[9] Concerning the destabilizing mutant made by removing 3 large hydrophobic residues, The authors should show that this protein is expressed in cells at similar levels to WT and also that it remains heat-modifiable in isolated membranes. If folding or OM insertion is an issue with this mutant, the force extension profiles will be misleading.

As we purify BtuB and its variants from the outer membrane and then reconstitute into liposomes, the protein concentration is constant for all our experiments (irrespective of levels of expression or insertion). BtuB3A has similar CD- and fluorescence emission spectra to that observed for the wild-type. However, the best evidence for insertion and folding of this variant, are the AFM data together with the *in vivo* data. The former data shows that the change in contour length between the first and second event (Lc1) is identical to that for wild-type, demonstrating that this variant forms a plugged lumen. Furthermore, our *in vivo* data shows that, if TonB-gating is prevented (by addition of excess TonB peptide), BtuB3A is resistant to the action of bactitracin – again indicating that the protein is fully folded and inserted.

[10] Supplementary Fig2, in Detailed BtuB purification section: the centrifugation steps are indicated with rpm without mentioning the rotor. It would be better to indicate the g forces.

The ×g for centrifugation steps that require a defined RCF are now stated in the re-written Methods section.

[11] Although not mentioned in this manuscript, the authors should really discuss the contribution of the proline-rich periplasmic domain of TonB to force propagation. What is known about this type of structure generating rigidity or force in general? What role might it play in the TBDT activation?

See response to Reviewer 2 [14].

Reviewer #2

[1] In all figures showing SMFS experiments the number of analyzed events should be given either directly in the figure or in the figure legend.

The requested data is now included in the relevant main and supplementary figure legends.

[2] In figure 6b only the outline of the frequency histograms is shown. I suggest the authors should show the histograms in a bar-representation (as they do in the preceding figures) and include the Gaussian functions fitting the distributions.

The amended histogram has been generated (Figure 6B).

[3] For readers who are not familiar with MD simulations the RMDS plots in Figure 5 C and D appear rather inconclusive. Also the figure legend lacks information in this context. As these plots show that the analyzed fraction of the protein maintains its unperturbed folded conformation this should be described accordingly. I would suggest also introducing the meaning of RMSD as a measure of distance between 2 structures (in this case perturbed and unperturbed).

RMSD is now defined in the Figure legend as “RMSD, a measure of the average deviation of the atomic positions from the experimental structure”. We also include the following sentence to explain the significance of these simulations: “The simulations show that the

RMSD of the PSD and the β -barrel are stable and of similar magnitude, demonstrating that the mechanically strong PSB is conformationally stable over this timescale” D) Equilibration of the system after 20 nm of unfolding by SMD.

[4] In Figure 5 A and B it appears that in the MD simulations the ligand remains stably bound to the extracellular cavity of BtuB, even after forced unfolding of the weak plug subdomain (a phenomenon we can confirm from similar simulations with FhuA). This could indicate that the mechanism requires more than just the mechanical opening of a transient channel and should be mentioned and discussed in the manuscript.

See reviewer 1 [6].

[5] Table 1 shows the result of an in vivo growth assay. How often was this assay performed? The number of replicates should be included in the figure legend.

The Table legend now states that “Two biological replicates were performed.”

[6] In the same assay the authors observed partial growth of colonies expressing the XL barrel mutation. Reduced growth indeed correlates with ~74% of the ButB molecules showing only one rupture event due to formation of a disulfide bridge (Figure 4). However, for a fully locked BtuB mutation one would expect to observe a null-growth phenotype. This could be achieved by repeating the growth assay in the presence of oxidizing agents such as CuSO₄.

The referee is correct in that performing the growth assay in an oxidizing agent should result in a null-growth phenotype. It is unclear, however, what additional insight such an experiment would provide. As recognised by the referee, our current in vitro AFM experiments (74 % showing one rupture event, in an oxidizing environment and 100 % double events in a reducing environment) on XL barrel are consistent with our *in vivo* assay (partial growth).

[7] Figure 6 C shows growth curves of bacteria. How often was this assay performed? Pls provide the statistics. Do the traces represent average curves of multiple experiments? As such growth assays sometimes show large variability between individual cultures of the same strain I suggest also including error bars/confidence intervals and statistical tests. Furthermore the number of replicates should be included in the figure legend.

The data in Figure 6C now includes error bars. The following sentence has been added to the legend: “Data points are averages of three technical repeats with error bars showing standard deviation.”

[8] The supplementary list of BtuB mutagenesis primers includes an entry “BtuB Y109C“. I cannot find such a mutation anywhere in the manuscript and suggest removal of the entry.

This entry has been deleted.

[9] The histogram shown in Fig 3D should be fitted with a multi Gaussian instead with three separate Gaussian functions. Particularly the Gaussians fitting the L1 and L2 distributions show a large overlap and thus considerable data sets are fitted (accounted) twice.

The referee is mistaken. As each unfolding event is discrete and clearly defined (i.e. L_c1 is the first observed and L_c2 is the second observed event), each is binned into their own dataset and used to generate histograms. These (**separate**) histograms are then plotted on the same axes. Whilst there is considerable overlap, each datum is accounted for once.

[10] Each histogram shown in Fig. 4 should be fitted with a multi Gaussian. As its done currently the data is not well fitted and the validity of the fits is questionable.

Please see point above. Futhermore, we note that the Gaussian distributions to L_{c1} and L_{c2} are provided as guides to the eye only. Quantitative data are derived only from Gaussian fits to ΔL_c histograms, which are all excellent. We think that the difference in the quality of the distributions of L_c and ΔL_c histograms are due to relatively high roughness of the surface, which is obviated upon calculation of each ΔL_c value.

[11] The authors interpret the rupture forces of 84-132 pN to be 'remarkably strong' compared to the low affinity. Such comparison must be done with great caution - as the pathway taken to mechanically rupture the bond may not the same as measured in affinity measurements. The authors should revise their statement accordingly.

We apologise for this lack of precision that arose during editing. This section has been re-written to make it clear that it is the off-rate that is being measured using force spectroscopy: “This is a remarkably strong interaction for a complex with a relatively low affinity (μM) if the association rate is assumed to be diffusion limited. For example, at similar loading rates, the dissociation of E9₂₀₋₆₆:Im9 ($K_d = 10^{-14}$ M, $k_{off} = 1.4 \times 10^{-6}$ s⁻¹)³³ and an antibody and its epitope ($K_d = 10^{-9}$ M, $k_{off} = 4.4 \times 10^{-3}$ s⁻¹)³⁴ occurs at a force of ~100 pN and ~160 pN respectively. These comparisons indicate that the TonB:Ton box complex is indeed mechanically robust and provide support for a slow association rate²⁹.”

[12] The authors write: "The lipid vesicles containing BtuB were then rolled onto a freshly cleaved mica surface and TonB Δ TMD immobilised to the AFM probe (Figure 1C)." It is not clear what 'rolled' means. Do the authors want to say that they adsorbed the lipid vesicles onto the mica? I have so far not been aware that vesicles can be rolled onto mica. In addition Fig. 1C shows a lipid bilayer on mica not a vesicle. Please revise to remove confusion.

The legend to Figure 1C has been amended to read: “and BtuB inserted into *E. coli* polar lipid extract liposomes and adsorbed onto a mica surface, is measured using an atomic force microscope. Note: adsorption of liposomes onto mica generates a heterogeneous surface with regions of single- and double-bilayer thicknesses.”

The “AFM cantilever derivatisation” section in Methods now reads: “For TonB:TB_{BtuB} dissociation, the peptide was attached to the AFM probe and TonB to the derivitised surface. For TonB:TB_{BDT} dissociation, TonB was attached to the AFM probe and the TB_{BDT} (inserted into a lipid bilayer) adsorbed to mica (see below).”

[13] The authors write: "We have demonstrated above that TonB Δ TMD:TB_{BtuB} is able to resist relatively high forces before dissociating." High is confusing here (and at some other passages of the text). Without a comparison high is meaningless. Forces around 100 pN may be rather small for others.

The “relatively high” was qualified by the following sentence in the submitted manuscript. To make this clearer we now state the following: “We have demonstrated above that TonB Δ TMD:TB_{BtuB} is able to resist levels of force that are sufficient to unfold protein domains with moderate mechanical strength^{35,36} at similar applied loading rates.”

[14] In the discussion the authors should clearly elaborate on which crucial aspects of the mechanism remains unanswered by their experimental setup and how the remaining questions may be addressed in the future. For example it is not clear how ligand transport can be linked to a mechanical pulling distance of 20 to 25 nm, particularly because it is not

known how the periplasmic lumen can undergo such movements. Thus it may rather be expected that not increasing the pulling distance per se will lead to the unfolding of the plug domain but rather a mechanical stress applied – such as in principle could be applied by force clamp measurements.

We have re-written the end of the first paragraph of the discussion to address this comment and similar comments regarding the total extension of the plug domain that is necessary (Reviewer 3 [5]) and the role of the proline-rich region of Ton B (Reviewer 1 [11]):

“The mechanisms of both inside-out energy transduction and the resultant vertical displacement of TonB by a distance similar to the width of the periplasm⁴³ are still unclear. Despite differences in their details, however, all the suggested models involve the formation of a force-transduction pathway which apply mechanical force to the plug domain. In these models, the vertical displacement of TB_{BtuB} that is required for channel opening may be a result of ‘passive’ processes (e.g. variations in the width of the periplasmic space⁴⁴ and /or the differential diffusion of the inner and outer membrane proteins⁴⁵). Alternatively, active processes dependent on the proton motive force may play a role. This includes a conformational change of the TonB linker resulting in a shortening in the end-to-end length of some part of the linker region (this work), by a transition of residues 66-100⁴⁶ from an extended polyproline type II to a shorter polyproline type I helix,¹⁴ or by the ExbBD-dependent^{23,47} rotary motion of TonB itself . ”

Reviewer #3:

[1] I find the presentation confusing at times. It may be that there is just a lot of data to present, but I found myself constantly checking what a given name meant or what a particular length should be. It might be a given that one has to read such a paper very carefully, but anything the authors’ can do to further improve clarity would be appreciated.

The referee is correct in stating there are a lot of data obtained from multiple experimental set-ups. We have looked carefully at our figures and read the text and feel that the suggested changes have improved the paper’s clarity.

[2] Page 7, paragraph beginning with “Closer examination...”: This entire paragraph is confusing. First, the latter value is 74 nm, yet it is said to be consistent with 47.2 nm; I assume the authors mean the former value? Also, it is not clear to me why TonB must also have a conformation 20.5 nm in length based on the former value (or the latter?). Finally, why is 24.5 nm expected?

We agree that this isn’t the clearest paragraph in the manuscript. The edited part now reads:

“The latter value is consistent with the sum of the end-to-end length of the PEG₂₄ linkers (19 nm) and the expected end-to-end length of the TonB_{ΔTMD}:TB_{BtuB} complex (5.5 nm) with an unstructured or unfolded 118-residue linker domain (47.2 nm, see Supplementary Note 2 and Supplementary Fig. 2C). The shorter modal L_c value (coincidentally also 47 nm) suggests that the linker domain of TonB_{ΔTMD} must also populate a force-resistant conformation approximately 22.5 nm in length (47 nm – 19 nm – 5.5 nm). To test this hypothesis, these experiments were repeated using TonB_{CTD} (i.e. the globular C-terminal domain without the linker). These experiments yielded a uni-modal distance-frequency histogram with a Gaussian fit value (22.4 nm) close to the predicted value (24.5 nm, the sum of the PEG₂₄ linkers (19 nm) and TonB_{ΔTMD}:TB_{BtuB} complex (5.5 nm), Supplementary Fig. 2B, Supplementary Note 3).”

[3] Finally, it seems like this paragraph comes too soon in the text; expected and average lengths don't come up until a few pages later (Fig. 3).

This section is discussing the minimal TonB_{ΔTMD}:TB_{BtuB} peptide system used to validate our approach (Figure 2) and text above this paragraph.

[4] Throughout: combined units are unclear. The meaning of “nms⁻¹” requires context to deduce. The authors should insert a dot between nm and s⁻¹ (for example) every time a combined unit appears.

All units are now presented in the same style (i.e. nm_s⁻¹)

[5] I would like the authors to speculate on how pulling over such large distances may be possible. The periplasm is only ~25 nm wide! Perhaps this fits with the fact that deltaLc usually ends up around 20-25 nm? But where is all the slack going? Does the TonB CTD end up touching the inner-membrane complex by the end?

This point has been addressed in the response to question [14] by reviewer 2.

[6] Can the authors address the TonB “dimer vs. monomer” debate with their results?

While there is evidence that TonB forms homodimers at some stage during transport (Postle et al. (2010) *Mbio* **1**, 307; Sauter et al. (2003) *J Bacteriol* **185**, 5747 and Ghosh & Postle, (2005) *Mol Microbiol* **55**, 276), it is accepted that TonB is monomeric when in contact with the receptor (Klebba (2016) *J Bacteriol* **198**, 1013 and Koedding et al. (2004) *J Biol Chem* **279**, 9978). The experiments described in the manuscript also focus on the unbinding of the Ton box peptide from the receptor and our results concur with the current paradigm: we see no difference in our results whether we use TonB_{CTD} (Supplementary Figure 3) or TonB_{ΔTMD} (Figure 3) to unfold the plug domain.

Minor issues

[7] In the introduction, the authors should cite the newly released structure of ExbB and ExbD

This work was published after submission of our manuscript. This is now reference 15.

[8] Page 7: “Closer examination of individual the” - “the” should come before “individual”.
Corrected.

[9] Figure 3 caption: “dissoicates” is misspelled.
Corrected.

[10] Page 10: “The Lc value is in accord...while the former value is close...” The authors should make clear that the “former value” is actually F_U.

Corrected. The sentence now reads: “The L_c value is in accord with the distance expected for the rupture of the complex without remodelling, while the F_U value is close to that observed for the dissociation of the minimal TB_{BtuB}:TonB complex observed previously at the same retraction velocity (1 μm s⁻¹, 113 ± 13 pN)”.

[11] Page 10: “Most importantly, the ΔLc-frequency histogram (Figure 4C grey histogram) yielded a significantly shorter modal value relative to wild-type (14 ± 3 nm and 20 ± 1 nm), confirming unfolding in the N-terminal portion of the plug.” I wouldn't say this is “significantly shorter”. Maybe just say it's 6 nm less?

The sentence now reads: “Most importantly, the ΔL_c -frequency histogram (Figure 4C grey histogram) yielded a modal value 6 nm shorter relative to wild-type (14 ± 3 nm and 20 ± 1 nm), confirming unfolding in the N-terminal portion of the plug.”

[12] Page 11: What is a “stringent concentration”? Maybe just say sufficient?

This sentence now reads “This strain requires vitamin B₁₂ to synthesise methionine and requires functional BtuB for full growth on minimal medium in the presence of 0.1-1 nM vitamin B₁₂.”

Reviewer #1 (Remarks to the Author)

The authors have addressed all of my concerns and the new manuscript is much improved.

Reviewer #2 (Remarks to the Author)

The authors have addressed all comments I had. The paper has been revised for improvement, which I think is great as it is.

Reviewer #3 (Remarks to the Author)

The authors have addressed all the reviewers' comments to my satisfaction.